# A novel male accessory gland peptide reduces female post-mating receptivity in the brown planthopper

Yi-Jie Zhang[1], Ning Zhang[1], Ruo-Tong Bu[1], Dick R. Nässel[2], Cong-Fen Gao[1]*, Shun-Fan Wu[1‡]*

1 Sanya Institute of Nanjing Agricultural University/College of Plant Protection, State Key Laboratory of Agricultural and Forestry Biosecurity, Nanjing Agricultural University, Nanjing, China, 2 Department of Zoology, Stockholm University, Stockholm, Sweden

‡ Lead contact
* gaocongfen@njau.edu.cn (CFG); wusf@njau.edu.cn (SFW)

## Abstract

Mating in insects commonly induces a profound change in the physiology and behavior of the female that serves to secure numerous and viable offspring, and to ensure paternity for the male by reducing receptivity of the female to further mating attempts. Here, we set out to characterize the post-mating response (PMR) in a pest insect, the brown planthopper *Nilaparvata lugens* and to identify a functional analog of sex peptide and/or other seminal fluid factors that contribute to the PMR in *Drosophila*. We find that *N. lugens* display a distinct PMR that lasts for about 4 days and includes a change in female behavior with decreased receptivity to males and increased oviposition. Extract from male accessory glands (MAG) injected into virgin females triggers a similar PMR, lasting about 24h. Since sex peptide does not exist in *N. lugens*, we screened for candidate mediators by performing a transcriptional and proteomics analysis of MAG extract. We identified a novel 51 amino acid peptide present only in the MAG and not in female *N. lugens*. This peptide, that we designate maccessin (macc), affects the female PMR. Females mated by males with *macc* knockdown display receptivity to wild type males in a second mating, which does not occur in controls. However, oviposition is not affected. Injection of recombinant macc reduces female receptivity, with no effect on oviposition. Thus, macc is an important seminal fluid peptide that affects the PMR of *N. lugens*. Our analysis suggests that the gene encoding the macc precursor is restricted to species closely related to *N. lugens*.

## Author summary

In insects, mating often induces a long-lasting change in the female behavior and physiology, called a post-mating response (PMR). This ensures numerous

**Data availability statement:** All of the RNA sequence data in this article have been deposited in the China National Center for Bioinformation database and are accessible in CRA019725. And the mass spectrometry proteomics data have been deposited to the Omix of the China National Center for Bioinformation database (https://ngdc.cncb.ac.cn/omix/) via the PRIDE partner repository with the dataset identifier OMIX007634.

**Funding:** This research was supported by the National Key Research and Development Program of China (2022YFD1700200) to SFW, the National Natural Science Foundation of China (No. 32022011 & 32472542) to SFW, the Guidance Foundation of the Sanya Institute of Nanjing Agricultural University (NAUSYMS15) to SFW and the Fundamental Research Funds for the Central Universities (No. KJJQ2024016) to SFW. The funders had no role in study design, data collection and analysis, decision to publish, or preparation of the manuscript.

**Competing interests:** The authors have declared that no competing interests exist.

and viable offspring, but also serves to secure paternity for the male by inhibiting the female receptivity to further mating attempts. Here, we demonstrate that a pest insect, the brown planthopper *Nilaparvata lugens*, also displays a PMR with decreased receptivity to further mating and increased egg laying. We furthermore find that seminal fluid extracted from the male accessory gland of *N. lugens* injected into females generates a PMR. Next, we identified a novel peptide unique to the male accessory gland (designated maccessin) and demonstrate that this peptide is responsible for the reduced receptivity in the PMR, but does not affect egg laying. The gene encoding maccessin appears unique to close relatives of *N. lugens*. This is similar to a *Drosophila* male accessory gland factor, sex peptide, which is known to induce a PMR, and occurs only in a limited number of *Drosophila* species.

## Introduction

Mating in insects commonly leads to a profound change in the physiology and behavior of the female that serves to secure a viable offspring and also to ensure paternity for the male by reducing receptivity of the female to further mating attempts [1–3]. This phenotypic switch has been especially well documented in *Drosophila* where the post-mating response (PMR) includes not only an increase in egg production, but also a reduced receptivity to courting males and alterations in feeding, metabolism, sleep pattern and others [1,3–10]. The trigger of this behavior switch is transferred from the male with the semen during copulation, and in *Drosophila* a major factor is a secreted 36 amino acid peptide, designated sex peptide [4,6,7,9]. This male-specific peptide, produced in the male accessory gland (MAG) acts primarily on a set of sensory neurons in the female reproductive tract known to express the sex-determination gene *fruitless* and connect to higher order brain circuitry consisting of *doublesex* expressing neurons [11–14]. Thus, transfer of sex peptide and activation of sex-specific neuronal circuits underlie part of the PMR in *Drosophila* females.

Interestingly, sex peptide and the related peptide DUP99B have only been identified in the genomes of a small set of *Drosophila* species and not in other insects [15]. The receptor for sex peptide [16] was found to be promiscuous and is additionally activated by myoinhibitory peptide (MIP), also known as allatostatin-B [17,18]. These authors suggested that MIPs are the ancestral ligands of the sex peptide receptor, but it is noteworthy that MIPs do not activate the PMR in *Drosophila* [17,18]. MIPs can be found in most insect species together with its receptor (MIPR). We henceforth collectively use MIPR for this receptor and the *Drosophila* sex peptide receptor. Although it is possible that MIPs could act as mediators of the PMR in insects that lack sex peptide, there is so far no evidence for this [19].

However, in some insects, it seems that the MIPR is involved in a portion of the PMR as a target of other hitherto unidentified ligands [19]. Examples are the oriental fruitflies *Bactrocera oleae* and *Bactrocera dorsalis* [20–22] and the cotton bollworm *Helicoverpa armigera* [23,24] where post-mating oviposition is affected by MIPR

knockdown. Diminishment of his receptor also affects oviposition in the Tobacco cutworm, *Spodoptera litura,* but has no effect on the PMR [25]. The identity of the authentic MIPR ligand(s) remains to be identified in these species.

In other species investigated, the MIPR seems not to be involved in the PMR. The mosquito *Aedes aegypti* is one such case [19]. Interestingly, however, a male-specific peptide was found in the *A. aegypti* MAG and shown to be transferred to females at mating [26]. This decapeptide, *Aedes* head peptide-1, does not act on the MIPR [26,27]. Instead the head peptide-1 receptor was identified as a short neuropeptide F receptor [28], and interestingly head peptide-1 induces a short refractoriness (less than 24 h) to insemination by other males [27], and other unknown factors control long-term refractory behavior [27]. Hence, the mosquito head peptide-1 and its receptor underlies a post-mating change in mate receptivity in females, but has no impact on fecundity, host-seeking or blood-feeding [27], suggesting that this peptide signaling is not fully equivalent to the sex peptide-MIPR axis in *Drosophila*. Finally, there is evidence for a non-peptidergic signal inducing PMR in the malaria mosquito *Anopheles gambiae* [29,30]. In this species a male-specific form of 20-hydroxyecdysone is sexually transferred to females to induce mating refractoriness [29].

We are interested in the molecular mechanisms and signaling pathway responsible for a possible PMR in a pest insect, the brown planthopper, *Nilaparvata lugens*. The mating behavior of *N. lugens* has been investigated in some detail [31–34] and a seminal fluid protein, Selenoprotein F-Like, has been identified that affects oviposition levels in mated *N. lugens* [35]. However, it is not yet clear whether females display a post-mating switch in further aspects of physiology and behavior. Our study identifies a distinct PMR in *N. lugens* with a change in female receptivity to males and an increase in oviposition, lasting for about 4 days. We found that extract from MAGs injected into virgin females induced a similar PMR, although lasting only for about 24h. Although the repertoire of seminal fluid proteins of *N. lugens* was already explored using transcriptomic and proteomic approaches [36], novel sequencing technology has been developed since then. Thus, the chromosome-level assembly of the brown planthopper genome with a characterized Y chromosome was reported [37] and further insect seminal fluid proteins have been reported in recent years [38,39]. Hence, we conducted a new screen for male accessory gland (MAG) seminal fluid proteins using improved sequencing technology and an updated version of the *N. lugens* genome. Our transcriptional and proteomics analysis of MAG extract identified a novel peptide precursor that is specific to the MAG in males and not found in female *N. lugens*. This was designated maccessin (macc). When exposing females to a second mating after first being mated with males where the (*macc*) gene was knocked down we did not observe any change in receptivity and injection of recombinant macc peptide reduces female receptivity. These manipulations did not affect oviposition. Thus, we propose that this novel MAG peptide is transferred via seminal fluid to females during copulation and induces a post-mating change in female behavior. It can be noted that we, and a previous study [40], identified another peptide precursor transcript in the MAG. This encodes an isoform (splice variant) of an ion transport peptide (ITPL-1). However, the same ITPL-1 peptide can also be produced by another splice variant in females [40]. We found that knockdown of this peptide in males and injection of recombinant ITPL-1 in females affected the female PMR similar to macc, but we cannot exclude that this peptide also acts as an endogenous factor released by cells in females.

## Results

### Brown planthopper females display a distinct post-mating response

Previous studies have described some aspects of the mating behavior of *N. lugens*. [31–34,41]. However, none of these studies elucidated the full sequence of this behavior in the *N. lugens*. Thus, we undertook a more detailed investigation of the courtship behavior in this species. During courtship in *N. lugens*, the males perform most of the behavioral steps while females only perform a few. The sequence of behaviors includes male abdominal vibration, virgin female abdominal vibration, then males following female, performing wing extension and abdominal vibration, followed by tapping, attempted copulation, copulation and termination of copulation (S1A-H Fig and S1 Video). There are no reports about a post-mating switch in female behavior and physiology in the *N. lugens*. Hence, we asked whether *N. lugens* females display a PMR

similar to that observed in *Drosophila* [1,6,7,42,43], malaria mosquito [29,30], Tephritid flies [44] and other insect species [45–47]. Indeed, we found that once a virgin female *N. lugens* has been mated, she is unwilling to accept another courting male (Fig 1A) and lays more eggs than virgin females (Fig 1B). The decreased receptivity of mated females is maintained for at least four days following copulation (Fig 1A). We confirmed that sperm was transferred into females after mating by analyzing the contents of the copulatory bursa of the female after mating (S2 Fig). Furthermore, we noted that the PMR in female *N. lugens* includes specific behaviors such as female abdominal vibration and extrusion of the ovipositor towards the courting males, which is also observed in the PMR of *Drosophila* [48,49] (S1I-S1J Fig and S2 Video). Our data, thus, show that female *N. lugens* exhibits a distinct PMR.

### Male accessory gland extract induces a post-mating response in *N. lugens*

Transfer of seminal fluid proteins into virgin females of different insect species, known to display a PMR, results in the repression of female sexual receptivity and stimulates their oviposition to levels similar to those of mated females [2,5,7,16,43,50–56]. However, in the Mexican fruit fly *Anastrepha ludens*, the stimulation of mating itself is sufficient to induce the female to lay an increased number of eggs laid, even if no ejaculate is received [57]. These authors also show that the reception of MAG fluid alone does not promote ovarian development and is not sufficient to increase the number of eggs laid [57]. Hence, we asked here whether seminal fluid proteins, transferred into female reproductive tract during copulation induces a PMR in *N. lugens*. seminal fluid proteins are primarily produced in the MAG and ejaculatory duct (Fig 1C) and transferred into female reproductive organs such as copulatory bursa and the spermatheca (Fig 1D) during copulation. There is only one spermatheca in the female reproductive system of *N. lugens*, and there is no ventral receptacle in the bursa copulatrix (Fig 1D).

We found that injection of extract from MAGs of *N. lugens* into abdomens of virgin females, affects receptivity to courting males (Fig 1E). These females displayed rejection behavior, e.g., abdominal vibration and ovipositor extrusion, typical of mated females (S3 Video). This effect lasts at least 24 hours after injection of MAG extract, and thus is shorter than the PMR seen after mating (Fig 1B and 1E). We observed that 36 hours after injection, virgins injected with MAG extract showed no significant difference in mating acceptance rates compared with virgins injected with the control solvent (Fig 1E). Since the receptivity rate of virgins after injection returned to normal levels at 36 hours, further long-term mating receptivity rates after injection were not studied. Our findings suggest that there could be other factors that play important roles in long-term PMR in the *N. lugens*. Another possibility is that relevant factors in the MAG extract need to be associated with (bound to) sperm to ensure gradual release of over a longer duration as was shown for sex peptide in *Drosophila* [58]. We furthermore observed that injection of MAG extract into virgin females stimulates egg-laying and leads to an increased percentage (20%) of females ovipositing, compared with solvent-injected controls (10%) (Fig 1F and 1G). These results are thus different from those observed in the Mexican fruit fly. In this fly, mated female increase oviposition without receiving an ejaculate [57]. Taken together, our findings show that *N. lugens* display a distinct PMR and that factors in MAG extract play an important role in this response.

### Transcriptome and proteome analysis of male accessory gland (MAG)

The seminal fluid protein genes and proteins of the brown planthopper have been investigated in a previous study [36]. However, improved sequence technology has been developed and a chromosome-level assembly of the *N. lugens* genome including a characterized Y chromosome was reported in 2021 [37]. Hence, to search for novel factors that may be transferred with the seminal fluid to induce a PMR, we performed a new transcriptional (RNA-seq quantification) and proteomics analyses of the MAGs from *N. lugens* (S3A Fig).

Illumina sequencing libraries were constructed by using mRNA from the MAG of *N. lugens*. A de novo transcriptome assembly was performed using Trinity v2.4.0 (min_kmer_cov:3, group_pairs_distance 500). We obtained 54,867,464.7 clean reads on average from samples of MAG (Fig 2 and S1 Table). After removing low-quality regions, adapters, and possible

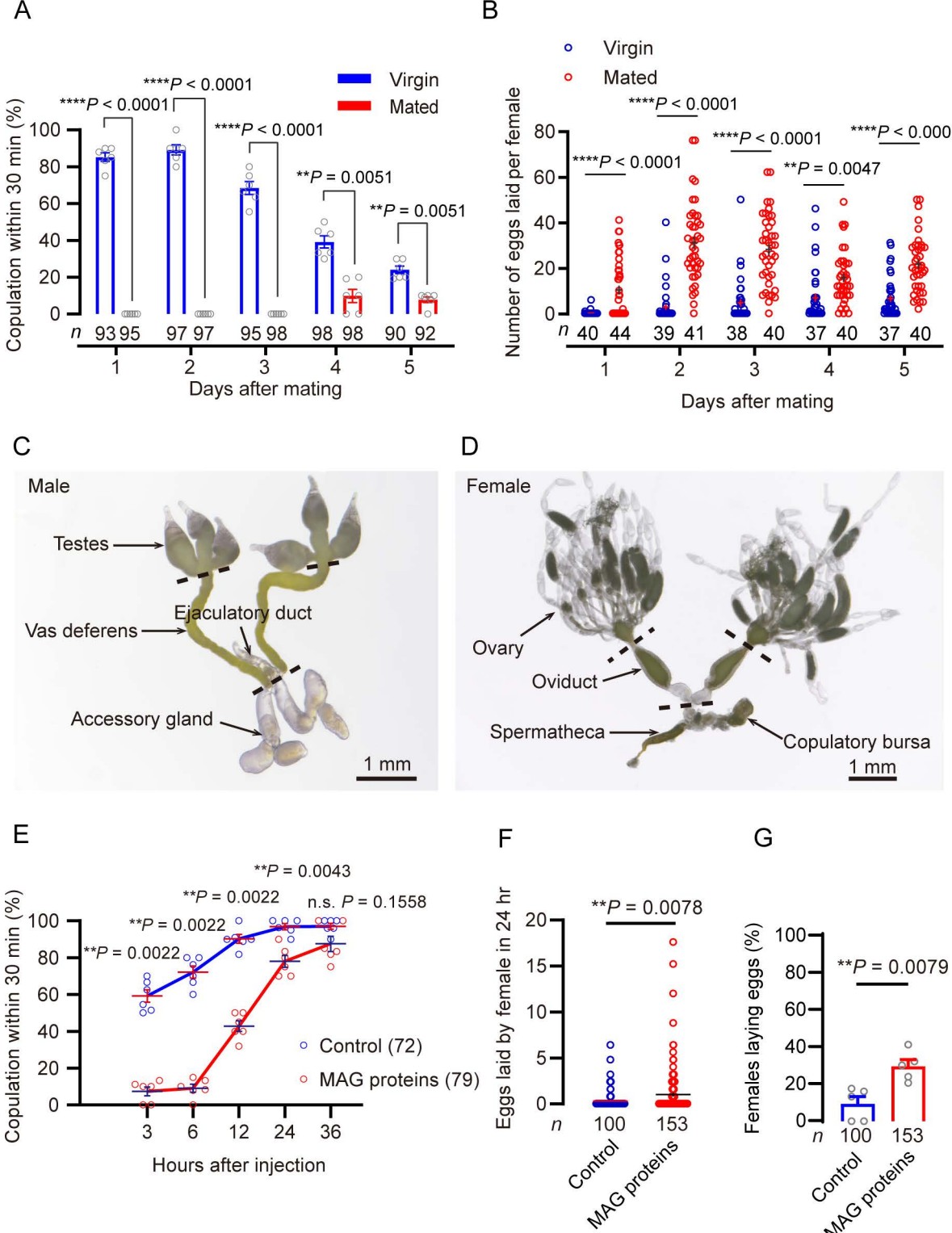

**Fig 1. Brown planthoppers display a distinct post mating response and male accessory gland extract induces a post-mating response.** A. Mating receptivity of virgin and mated females. Graph shows the proportion of females that accept males as virgins and as mated insects over five days. Rates of receptivity differs significantly between mated and virgin females on each of the initial four days. The numbers below the bars denote

total number of animals. Error bars indicate SEM throughout. **$P<0.01$, ***$P<0.001$, and ****$P<0.0001$, two-way repeated measures ANOVA followed by holm-šídák's multiple comparisons test. B. Numbers of eggs laid per female. Note that wild type *N. lugens* females are known to mate less than six days after eclosion in our lab based on a previous study [31]. The small circles denote the number of eggs laid by an individual female, and the numbers below the bars denote total number of animals. Error bars indicate SEM throughout. *$P<0.05$, **$P<0.01$, ***$P<0.001$, ****$P<0.0001$, and ns (non-significant), $P>0.05$, two-way repeated measures ANOVA followed by holm-šídák's multiple comparisons test. C. The reproductive system of the male brown planthopper includes the testes, vas deferens, accessory glands and ejaculatory ducts. D. The reproductive tract of the female brown planthopper includes the ovary, oviduct, spermatheca and copulatory bursa. E. Receptivity to mating in virgin females after injection of male accessory gland proteins extracted from the male *N. lugens*, measured as percentage of females that copulated within 30 min. The number of refractory females is high 3-6 h after MAG extract injection compared with saline injection, but then declines. The numbers in brackets denote total number of animals. *$P<0.05$, **$P<0.01$, and ns (non-significant), $P>0.05$, for comparisons against control; Mann–Whitney test. F. Numbers of eggs laid per virgin female in 24 h after MAG extract injection. The small circles denote the number of eggs laid by an individual female, and the numbers below the bars denote total number of animals. **$P<0.01$, Student' t-test. G. Percentage of virgins laying eggs during 24 h after MAG extract injection. The small circles denote the number of biological replicates and the numbers below the bars denote total number of animals. ***$P<0.001$, Mann–Whitney test (Five biological replicates were performed with at least ten insects per replicate for each experiment).

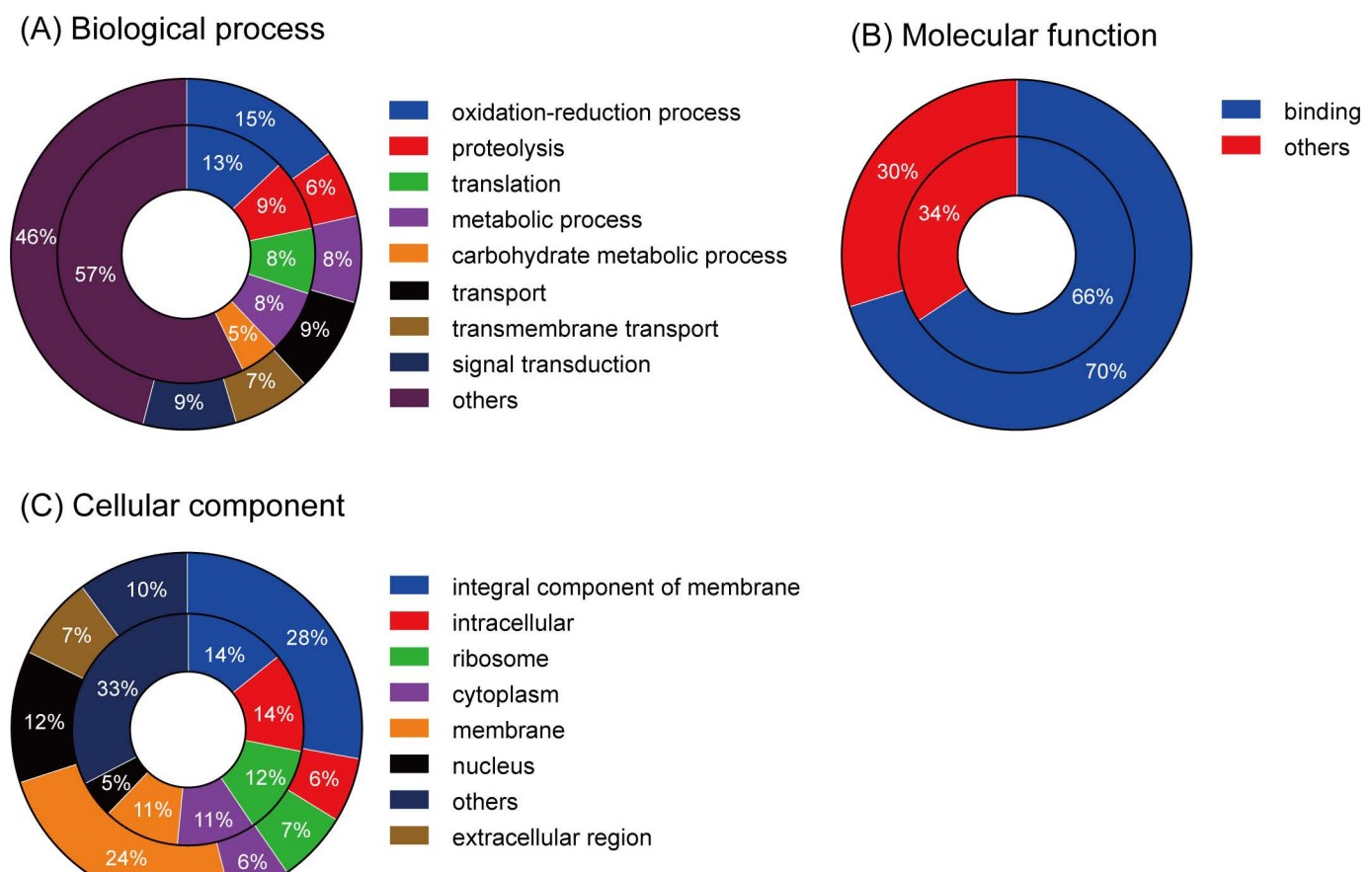

**Fig 2. GO classification of brown planthopper MAG transcripts and proteins.** Genes and proteins are classified according to gene ontology annotations, and the proportions of each category are shown in terms of percentages of (A) biological processes, (B) molecular functions, and (C) cell components. We grouped all the GO terms with less than, e.g., 5% and term them "others" in A, B and C panels. Outer ring represents transcriptome GO analysis and inner ring represents proteome GO analysis.

contamination using NGSQC software [59], we obtained more than 6 giga base clean bases with a Q20 percentage over 98%, Q30 percentage over 94%, and a GC percentage between 38.94 and 41.58% (S1 Table). After alignment by Bowtie, 61.01–65.66% and 61.97–66.21% unique reads were mapped into the reference genome of *N. lugens* [37]. All of the RNA sequence data in this article have been deposited in the China National Center for Bioinformation database and are accessible in CRA019725. To identify the putative function of assembled transcripts, sequence similarity search was conducted against the NCBI non-redundant (NR) and Swiss-Prot protein databases using BLASTx search with a cut-off E value of $10^{-5}$.

Proteomic analysis was performed using Label-free quantitative LC-MS/MS proteomics method [60,61]. Production data was searched against *N. lugens* MAG transcriptome databases using the Proteome Discoverer 2.2 (PD2.2, Thermo) and identical search parameters. The mass spectrometry proteomics data have been deposited to the Omix of the China National Center for Bioinformation database (https://ngdc.cncb.ac.cn/omix/) via the PRIDE partner repository with the dataset identifier OMIX007634.

Proteogenomic analysis of the assembled *N. lugens* MAG transcriptome revealed 214,461 transcripts corresponding to 27,118 genes, with transcript lengths ranging from 201 bp to 17,079 bp. Subsequent proteomic profiling identified 366,540 total spectra, of which 101,639 (27.73%) passed quality control thresholds. These validated spectra enabled the identification of 28,998 unique peptides and 3,080 high-confidence proteins. Notably, 11.36% of the assembled transcripts demonstrated evidence of protein translation, establishing a direct link between the transcriptomic and proteomic datasets. This integrated approach provides comprehensive molecular evidence spanning from gene prediction to protein verification in the *N. lugens* MAG system.

Gene ontology (GO), an international standardized gene functional classification system, was used to classify the function of the predicted *N. lugens*. genes using GOseq (v1.10.0) [62] and InterProScan 5 to GO analysis for proteome [63]. Based on sequence homology, a total of 16,367 transcripts (36.78%) and 1911 proteins could be categorized into three main categories: biological process, cellular component, and molecular function, with 112 function groups in the transcripts and 112 in the proteins (Fig 2). Genes and proteins involved in oxidation-reduction process was the largest category in biological processes, including 1121 (15.3%) transcripts, 113 (12.9%) proteins. There were 459 (6.3%) transcripts and 78 (8.9%) proteins involved in proteolysis, 72 proteins involved in translation and 69 involved in metabolic process, and the number of proteins involved in signal transduction and transport reached 630 and 644 transcripts, respectively (Fig 2A). In the cellular component category, proteins involved in integral component of membrane (1750, 27.7% transcripts and 79, 14.2% proteins respectively) and membrane (1509, 23.9% transcripts and 59, 10.6% proteins) were all prominently represented (Fig 2C). The genes and proteins associated with 'binding' were 70.2% and 65.7% respectively in the molecular function category (Fig 2B). This pattern of distribution is typically seen in the transcriptome of samples undergoing development processes [64]. In our database, 586 transcripts were annotated as related to metabolic processes, which suggests that this analysis provides abundant information on novel genes involved in metabolic pathways, including secondary metabolism.

The Kyoto Encyclopedia of Genes and Genomes (KEGG) database was utilized to categorize gene function and pathways. There were 10,582 transcripts mapped into 229 KEGG pathways. The maps with the highest transcripts representation are signal transduction (1565 transcripts, 14.8%), followed by endocrine system (942 transcripts, 8.9%), carbohydrate metabolism (795 transcripts, 7.5%), and amino acid metabolism (584 transcripts, 5.5%) (S3A Fig). The presence of abundant metabolic pathways has also been found in the proteomics of accessory gland of brown planthopper [36]. There are 1047 proteins that mapped into 122 KEGG pathways, with the highest protein representation in global and overview maps (344, 16.5%), followed by carbohydrate metabolism (228, 10.9%), transport and catabolism (202, 9.7%), folding, sorting and degradation (202, 9.7%), translation (195, 9.3%), overview (160, 7.7%), amino acid metabolism (122, 5.8%) (S4B Fig). Overview is a second-level entry in the KEGG category, which is grouped under the first-level entry Metabolism and contains the following third-level entries: Carbon metabolism (ko01200), Biosynthesis of amino acids (ko01230), Fatty acid metabolism (ko01212) and 2-Oxocarboxylic acid metabolism (ko01210). Global and overview maps is also a second-level

entry in the KEGG category, which is grouped under the first-level entry Metabolism and contains the third-level entries Metabolic pathways (ko01100). The abundant metabolic pathways found in both the transcriptome and proteome suggest that the sample has a complex and active metabolic network, which is crucial for its development and functions to the *N. lugens*. The specific distribution of transcripts and proteins in different pathways can provide insights into the functional characteristics and regulatory mechanisms of the organism's biological processes.

## Identification of seminal fluid proteins of *N. lugens*

Genes encoding seminal fluid proteins were predicted using both the *N. lugens* transcriptome assembly and proteomic analysis. The screening criteria for seminal fluid proteins in the brown planthopper (*N. lugens*) are as follows:

1. Gene Annotation: Genes labeled as "seminal fluid protein" in previous studies were selected through screening gene annotations.

2. Gene Expression Level: The gene expression levels in the MAG (metagenome - assembled genome) were recorded and ranked from high to low.

3. Signal Peptide Presence: Genes were further checked for the presence of signal peptides.

4. Candidate Selection: Genes that have high expression levels in the MAG and also possess signal peptides, among the "seminal fluid protein" genes, are considered as candidate genes for seminal fluid proteins.

We identified a total of 373 putative seminal fluid proteins from the MAG transcriptome data. Of these, 209 sequences were confirmed by proteomic analysis (S3B Fig and S2 Table). Among of these, 131 putative seminal fluid proteins have signal peptides and are likely to be secreted by the MAG (S2 Table). One of these gene transcripts in the MAG encodes an ion transport peptide-like peptide precursor (ITPL-1) (S2 Table). It had previously been reported that one splice isoform (*itpl-1*) of this gene is specifically expressed in the MAG of *N. lugens* [36,40]. However, the same mature peptide (ITPL) can be produced from another three splice variants of the same gene in different tissues of females [36,40], suggesting that this ITPL-1 peptide is also endogenous to females.

Based on gene annotation and functional analysis, we prioritized seminal fluid protein candidate genes involved in signal transduction due to their potential biological significance. Among these, an ion transport peptide precursor exhibited exceptionally high expression levels in the MAG and contained predicted signal peptides, qualifying them as high-confidence seminal fluid protein candidates (S2 Table). While functional testing of several highly expressed seminal fluid protein candidates revealed no significant impact on female re-mating behavior (S5 Fig), transcriptomic and mass spectrometric analyses identified a peptide precursor in the MAG annotated as a signal transduction molecule. This gene, distinct from the tested candidates, showed remarkably high MAG-specific expression (S2 Table), suggesting a potential novel role in post-mating responses that warrants further investigation. This is encoded by the gene BAO00947, which has been previously annotated in *N. lugens* as a peptide precursor, but in whole animal analysis (not specifically MAG) [64]. Amino acid sequence analysis revealed that its first 19 amino acids were predicted to be a signal peptide. It may be a secreted protein that is likely to be transferred to the female during mating. The peptide encoded on this precursor is 91 amino acids and the mature peptide between the KR-cleavage sites is 51 amino acids long, and has six cysteines that can form three disulfide bridges, spaced in a fashion resembling ion transport peptides [65,66]. There is no C-terminal amidation signal, suggesting that the peptide is non-amidated. We designate this peptide maccessin (male accessory gland peptide; macc). In Fig 3A, we show the amino acid sequence of the predicted neuropeptide precursor encoded by BAO00947, including the signal peptide and cleavage sites. A macc peptide fragment (ATLGEYTY) could also be detected in MAG tissue extract in our proteomics analysis (S2 Table). We performed a semi-quantitative RT-PCR analysis of *macc* and found that it is only expressed in the MAG, and cannot be found in males with the MAG removed, or in females (Fig 3B). Thus, *macc* is a male and tissue-specific peptide.

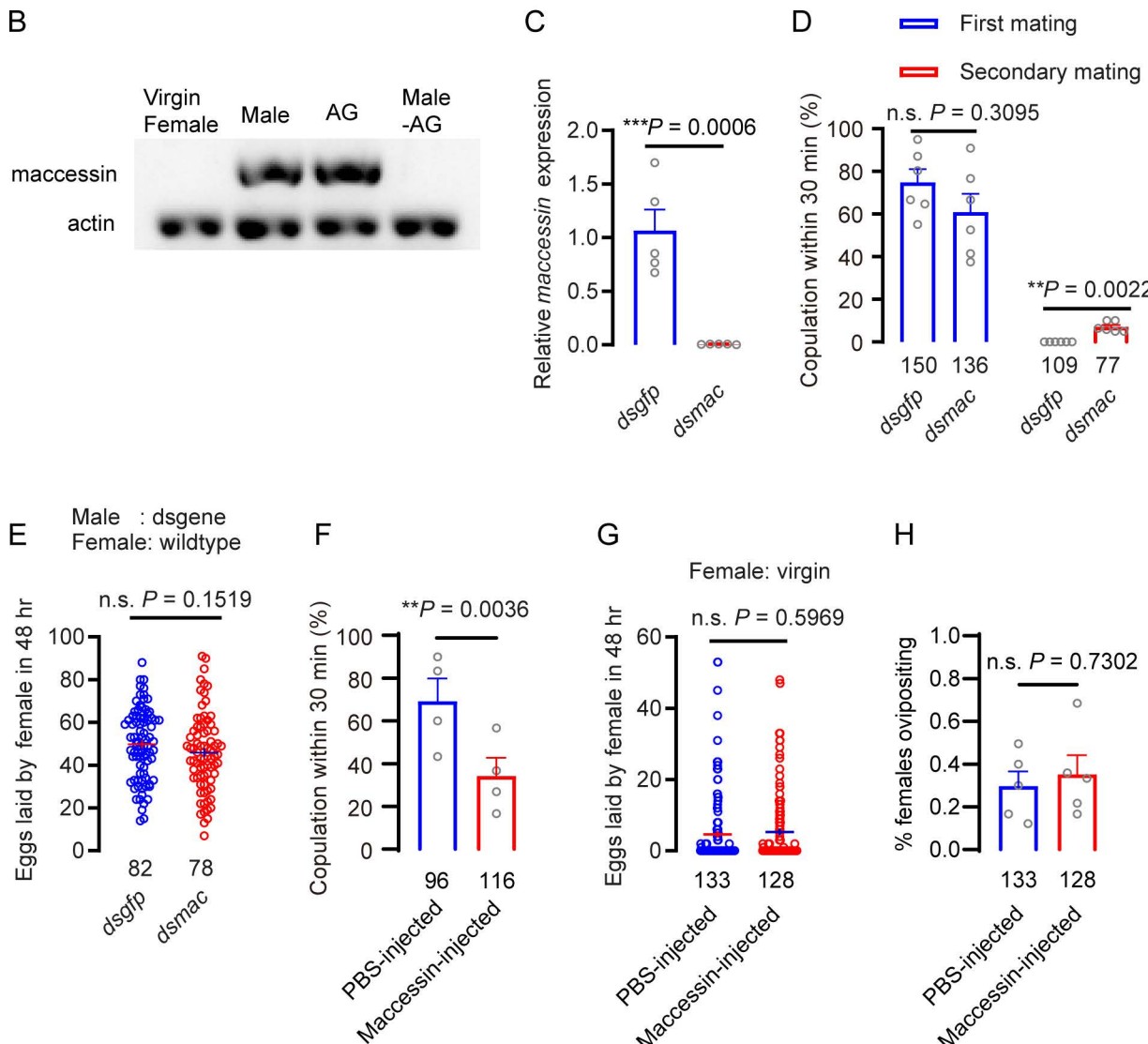

**Fig 3. Maccessin reduces female receptivity but not oviposition in brown planthopper.** **A** Amino acid sequence of the maccessin precursor in the brown planthopper. Yellow marker indicates signal peptide sequence. The six cysteines (blue marker) can form three disulfide bonds. The mature peptide is shown in gray background. The red-marked KR sites indicate cleavage sites for mature peptide. Note that there is no C-terminal amidation signal. The amino acids marked in red font indicate the peptide detected by mass spectrometry. **B.** The tissue distribution of *maccessin* analyzed by semi-quantitative RT-PCR. Whole animals (virgin female, male), male accessory glands (AG) and whole males with accessory glands removed (Male – AG), were assayed. **C.** The gene-silencing efficacy of *Maccessin* in male insects following dsRNA injection (three days after injection which corresponds to two days after eclosion) assayed by qPCR. The small circles denote the number of replicates. Data are shown as mean±s.e.m. Mann–Whitney test. ***$P < 0.001$. **D.** Receptivity of virgin and mated females, scored as the percentage of females that copulated within 30 min. The small circles denote the number of replicates; the numbers below the bars denote total number of animals. Data are shown as mean±s.e.m. **$P < 0.01$, and ns (non-significant),

*P* > 0.05, two-way repeated measures ANOVA followed by holm-šídák's multiple comparisons test. **E.** Number of eggs laid per female in 48 h. The dsRNA was injected in males at the end of the fifth instar, and the first mating took place after two days of eclosion. The numbers below the bars denote total number of animals. At least four biological replicates with at least ten insects per replicate for each experiment. Data are shown as mean ± s.e.m. Student's t-test. ns (non-significant), *P* > 0.05. **F.** Percentage of virgins mating during 6 h after injection. Each virgin was injected with 30 nL of 1 × PBS or 100 µmol/L maccessin. The small circles denote the number of replicates; the numbers below the bars denote total number of animals. Data are shown as mean ± s.e.m. Ns (non-significant), *P* > 0.05, for comparisons against PBS; Mann–Whitney test. **G.** Number of eggs laid after injection of maccessin in virgin females. Data are shown as mean ± s.e.m. Student's *t*-test. ns (not significant), *P* > 0.05, for comparisons against *t* PBS injected. The numbers below the bars denote total number of animals. At least four biological replicates with at least ten insects per replicate for each experiment. Each virgin was injected with 30 nl of 1 × PBS or 100 µmol/L maccessin. **H.** Percentage of virgins laying eggs during 48 h after maccessin injection. The numbers below the bars denote total number of animals. At least four biological replicates with at least ten insects per replicate for each experiment. Each virgin was injected with 30 nl of 1 × PBS or 100 µmol/L maccessin. Mann–Whitney test, ns (not significant), *P* > 0.05, for comparisons against PBS injection.

The *macc* gene could not be found in related insect species, such as small brown rice planthopper, *Laodelphax striatellus*, and whitebacked planthopper, *Sogatella furcifera* or other more distantly related insects including *Drosophila* and mosquitos (S6A Fig). However, we found a gene homologous to *macc* in the closely related species *Nilaparvata muiri*. Thus, the macc peptide is well conserved between *Nilaparvata muiri* and *Nilaparvata lugens* (S6B Fig).

### The novel peptide maccessin reduces female receptivity but does not induce oviposition in *N. lugens*

Next, we asked whether the novel peptide macc plays a role in the PMR of *N. lugens*. We diminished the expression of the *macc* gene in males by injection of dsRNA with a knockdown efficiency of more than 90% at three days after injection (Fig 3C), and found that a significant number of females that had been mated with these males would mate again with wild type males (*P* < 0.05) (Fig 3D). Such re-mating is never observed in the control (*dsgfp*-injected) group (Fig 3D). The number of eggs laid per female displayed no difference between females mated with *dsgfp*- and *dsmacc*- injected males (Fig 3E).

If macc is transferred from a male to a female during copulation it could enforce his paternity by reducing receptivity of the female to further mating attempts. A similar response should be seen after macc peptide injection into a virgin female. To test this, we generated recombinant macc for injections. Next, we injected individual wild-type virgin females with either buffer or mature macc peptide and allowed them to recover for 6 hr in groups. This extended recovery time was required because virgins tested shortly after injection did not mate regardless of the substance injected. After recovery, injected females were exposed to wild-type males for 30 min, an exposure time that was sufficient for nearly all control females to show receptivity. We found that injection of macc significantly reduces virgin female receptivity (Fig 3F). Besides this, we observed an obvious post mating response in virgin females 6 hours after injection with macc, such as ovipositor extrusion, which is never seen in PBS injected females. However, injection of macc did not induce oviposition in virgin females (Fig 3G and 3H). Furthermore, we selected eight other seminal fluid protein candidate genes with high expression levels for functional testing (dsRNA injections), and found that none of them had a significant effect on post-mating response (S5 Fig). In summary, our data show that females mated with males with diminished *macc* will mate again and that injection of macc in females reduces receptivity, but does not induce increased oviposition in virgin *N. lugens*.

### An ITP-like peptide (ITPL-1) also reduces female receptivity but does not induce oviposition in *N. lugens*

Previous work identified an ITP precursor gene (Accession number: XP_039277955) in the brown planthopper that gives rise to an amidated ITP peptide (ITPa) and four distinct ITPL transcripts (ITPL-1–4) containing different 5' UTRs [36]. The open reading frame (ORF) and the 3' UTR regions of the four ITPL transcripts are identical and encode the same non-amidated ITPL peptide (S7 Fig) [36,40]. The four ITPL transcripts display differential spatio-temporal expression patterns, where ITPL-1 is exclusively expressed in males, and specifically only in the male reproductive system [40]. However, the three other splice forms *itpl-2*–4 are expressed in other tissues in both males and females [40]. *Itpl-2* and *itpl-3*

are widely expressed in the integument, brain, the male or female reproductive system, gut (including Malpighian tubules) and fat body. *Itpl-4* is expressed in the brain and integument. We confirmed these findings by RT-PCR and qPCR and found that *itpl-1* is exclusively expressed in the MAG and not in females (S8A-C Fig).

As noted above, the mature ITPL peptide that can be generated from the *itpl*-1 transcript is identical to the ones derived from *itpl-2–4*, suggesting that the ITPL-1 peptide can be produced also in females. Nevertheless, we next asked whether male-derived ITPL-1 plays a role in the PMR of *N. lugens*. Thus, we injected dsRNA that target *itp/itpl* (it is not possible to only silence *itp*) or only *itpl* (dsRNA to target exon 3) to determine whether peptide-deficient males can induce a PMR in female *N. lugens*. Our data show that dsRNA injection significantly reduced the transcript levels of *itp* and the four splice forms *itpl1–4* in the brown planthopper (S8D-G Fig). Like for *macc* knockdown, we observed that a number of females who first had mated with *dsitp/itpl* and *dsitpl* males did remate with wildtype males, which is not seen after a first mating with the *dsgfp* injected control males (Fig 4A). However, this remating rate is lower than 20 percent of the females mated in the secondary mating (Fig 4A). The number of eggs laid per female displayed no difference between females mated with *dsgfp*, *dsitp/itpl* and *dsitpl* injected males (Fig 4B). To specifically silence the *itpl-1* isoform, we synthesized small interfering RNAs (siRNAs), which target the exon 1a (S7 Fig). We again observed that a number of females who first mated with itpl-1 siRNA males did remate with wild type males, which was not seen in controls (Fig 4C). However, the number of eggs laid was not changed after copulating with *itpl-1* siRNA males (Fig 4D). In summary, females who mated with males with *itpl-1* knockdown displayed remating after first mating (similar to knockdown of the novel MAG peptide *macc,* but with less potency).

Next, we used recombinant amidated ITP (ITPa) and non-amidated ITPL-1 injection in females to test the effect on the PMR. Individual wild-type virgin females were injected with either buffer, mature ITPa or ITPL-1 peptide. We found that injection of ITPL-1 reduced the receptivity of virgin females (Fig 4E). However, ITPa only has a weak (non-significant) effect on receptivity of virgin females (Fig 4E). We hypothesize that since ITPa is not produced in the MAG, any effect of injected peptide on female receptivity would reflect the action of endogenous female ITPa in post-mating behavior. Furthermore, since peptides identical to ITPL-1 (ITPL-2–4) are likely to be produced endogenously in females, we cannot exclude that injected ITPL-1 also mimics endogenous peptide, at least partly. We did not find that oviposition increased after ITPa or ITPL injection into virgin females (Fig 4F and 4G).

## Myoinhibitory peptides (MIPs) and *Drosophila* sex peptide do not trigger a post mating response in *N. lugens*

We found the seminal fluid-derived peptide, macc, can trigger a PMR in the female brown planthopper, but since it is a novel peptide unrelated to previously known ones the receptor is unknown. Thus, we wondered whether the MIPR previously implicated in *Drosophila* and other insects may act as a receptor of macc. However, first, we asked whether the known MIPR ligands sex peptide or MIPs play any role in the PMR of *N. lugens*. We thus tested whether *Drosophila* sex peptide can induce a PMR in brown planthopper. As a control, we showed that injection of sex peptide significantly inhibits receptivity of virgin female *Drosophila* (Fig 5A). However, injection of *Drosophila* sex peptide does not diminish receptivity of virgin *N. lugens* (Fig 5B).

MIPs have been reported as the ancestral ligands of the promiscuous *Drosophila* sex peptide receptor (also known as MIPR) [17,18,67,68], but do not induce a PMR in *Drosophila* [17,18]. Since MIP signaling is ubiquitously present in insects [17,18] and the MIPR has been implicated in the PMR in a few insect species [see [19]], we asked whether MIP signaling is involved in the PMR in *N. lugens*. First, we cloned the *mip* gene of the *N. lugens* and found that it encodes four mature MIP peptides, MIP1 - MIP4 (S9 Fig). Of these, MIP2 is predominantly expressed in *N. lugens* with eight paracopies in the precursor (S9 Fig). Next, we examined the expression pattern of *mip* in *N. lugens*. Investigating different developmental stages by real-time PCR, we found that *mip* transcript levels are boosted in third instar larvae and adult males. Transcripts are more abundant in adult males than in females. Of the different tissues, *mip* was detected in highest levels in the head of both male and female *N. lugens* (S10A and S10B Fig).

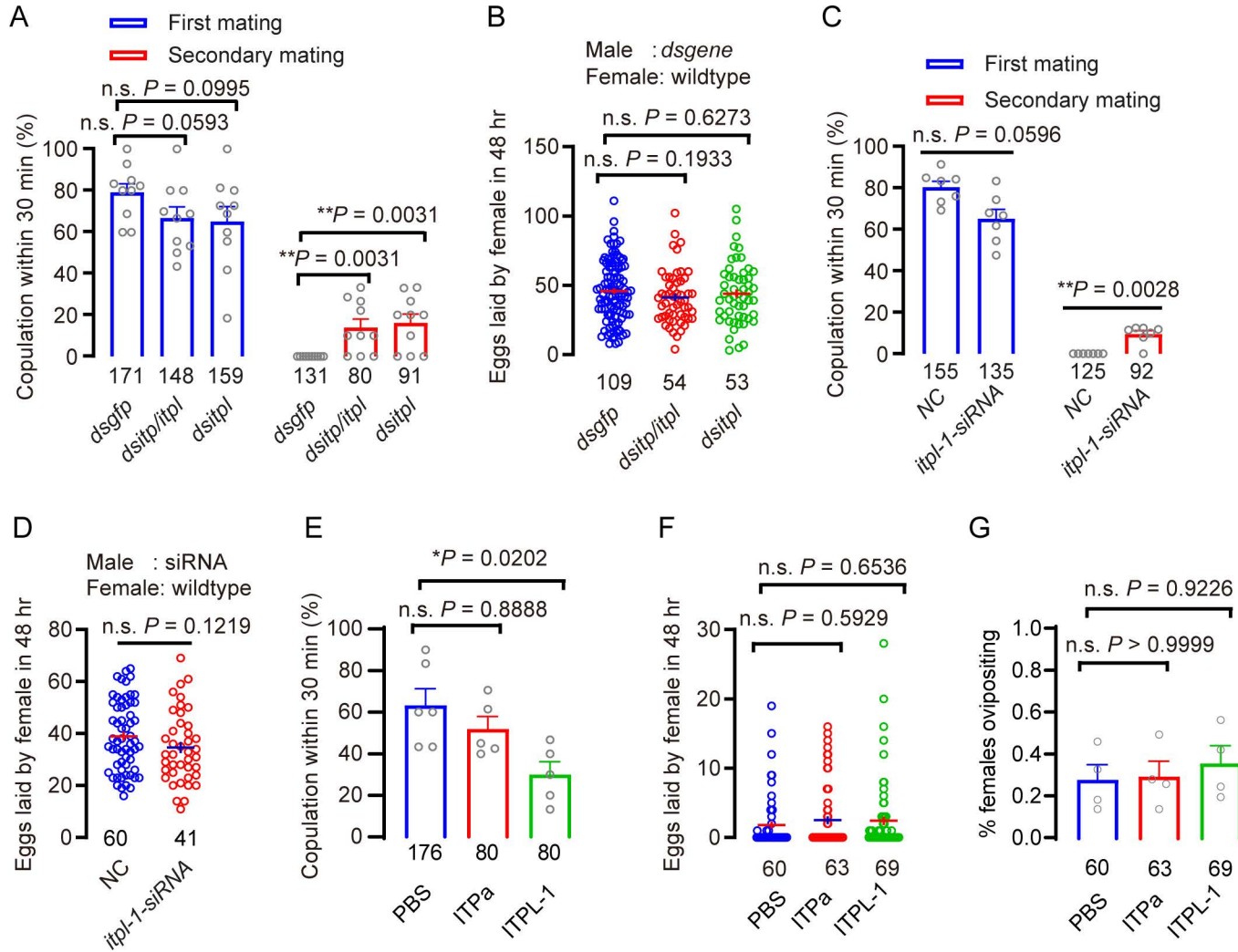

**Fig 4. ITPL-1 also reduces female receptivity but does not affect oviposition in the brown planthopper.** A. Receptivity of virgin and mated females after *itp/itpl* and *itpl* knockdown by dsRNA injections, scored as the percentage of females that copulated within 30 min. The small circles denote the number of replicates; the numbers below the bars denote total number of animals. Data are shown as mean ± s.e.m. **$P < 0.01$, and ns (non-significant), $P > 0.05$, two-way repeated measures ANOVA followed by holm-šídák's multiple comparisons test. B. Number of eggs laid per female in 48 h. dsRNA was injected in males at the end of the fifth instar, and the first mating took place two days after eclosion. The small circles and the numbers below the bars denote total number of animals. Data are shown as mean ± s.e.m. Student's t-test. ns (not significant), $P > 0.05$, for comparisons against *dsgfp* injected. C. The mating receptivity rates of female to *NC (negative control)* and *Nlitpl1-siRNA* injected male courtship. The *Nlitpl1-siRNA* is designed to target a sequence specific for *Nlitpl1*, differentiating it from the other four spliceosome components. The small circles denote the number of replicates; the numbers below the bars denote total number of animals. Data are shown as mean ± s.e.m. **$P < 0.01$, and ns (non-significant), $P > 0.05$, two-way repeated measures ANOVA followed by holm-šídák's multiple comparisons test. D. Number of eggs laid per female in 48 h. The experimental protocol and symbols are the same as [Fig 4B]. The small circles and the numbers below the bars denote total number of animals. Data are shown as mean ± s.e.m. Student's t-test. ns (not significant), $P > 0.05$, for comparisons against *NC* injected. E. Receptivity of virgin females after PBS or peptide injection, scored as the percentage of females that copulated within 30 min. Each virgin was injected with 30 nl of 1 × PBS, 400 μmol/L ITP or 400 μmol/L ITPL. The numbers below the bars denote total number of animals. Data are shown as mean ± s.e.m. Groups that share at least one letter are statistically indistinguishable; Kruskal–Wallis test followed by Dunn's multiple comparisons test with $P < 0.05$. F. Number of eggs laid after injection of ITPa or ITPL-1 peptide in virgin females. The small circles denote the number of eggs laid by an individual female, and the numbers below the bars denote total number of animals. Data are shown as mean ± s.e.m. Groups that share at least one letter are statistically indistinguishable; Kruskal–Wallis test followed by Dunn's multiple comparisons test with $P < 0.05$. G. Percentage of virgins laying eggs during 48 h after ITP or ITPL injection. The numbers below the bars denote total number of animals. The small circles denote the number of eggs laid by an individual female, and the numbers below the bars denote total number of animals. Data are shown as mean ± s.e.m. Groups that share at least one letter are statistically indistinguishable; Kruskal–Wallis test followed by Dunn's multiple comparisons test with $P < 0.05$.

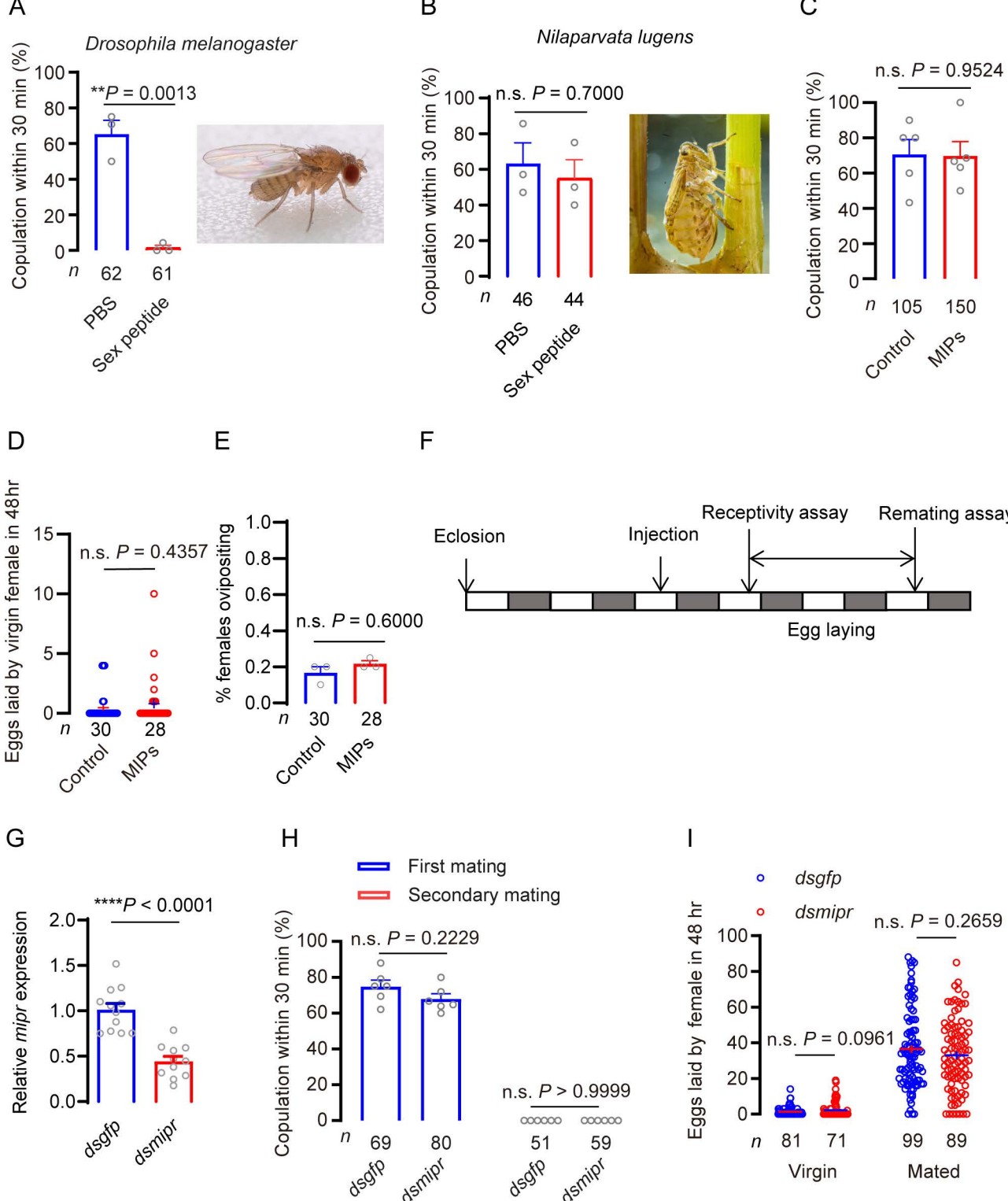

**Fig 5. MIP and MIPR are not involved in the post-mating response of brown planthoppers.** A. Receptivity of virgin female fruit flies after PBS or sex peptide injection, scored as the percentage of females that copulated within 30 min. The small circles denote the number of replicates; the numbers below the bars denote total number of animals. Data are shown as mean±s.e.m. Mann–Whitney test. **P<0.01, for comparisons against PBS injected.

B. Receptivity of virgin female *N. lugens* after PBS or sex peptide injection, scored as the percentage of females that copulated within 30 min. Each virgin was injected with 30 nl of 600 µmol/L sex peptide. Data are shown as mean ± s.e.m. Mann–Whitney test. ns: no significant, $P > 0.05$, for comparisons against PBS injected. C. Receptivity of virgin female *N. lugens* after injection of PBS or MIP mixture (MIPs), scored as the percentage of females that copulated within 30 min. Each virgin was injected with 30 nl of 1 × PBS or 600 µmol/L MIPs. MIPs are a blend of MIP1, MIP2, MIP3 and MIP4. The small circles denote the number of replicates; the numbers below the bars denote total number of animals. Data are shown as mean ± s.e.m. Mann–Whitney test. ns: no significant, $P > 0.05$, for comparisons against PBS injected. D. Number of eggs laid after injection of MIPs in virgin females. The small circles denote the number of eggs laid by an individual female. Data are shown as mean ± s.e.m. Ns: no significant, $P > 0.05$, for comparisons against PBS injected. E. Percentage of virgins laying eggs during 48 h after MIP injection. The numbers below the bars denote total number of animals. Data are shown as mean ± s.e.m. Data are shown as mean ± s.e.m. Mann–Whitney test. ns: no significant, $P > 0.05$, for comparisons against PBS injected. F. Protocol for behavioral experiments in G and H. G. Relative expression of *mipr* transcript in females injected with dsRNA. Females were collected immediately after the re-mating receptivity experiment, and total RNA was extracted to monitor the relative expression of *mipr*. Data are shown as mean ± s.e.m. ****$P < 0.0001$; Student's t test. The small circles denote the replicates. H. Receptivity of virgin and mated females, scored as the percentage of females that copulated within 30 min. Data are shown as mean ± s.e.m. Ns (not significant), $P > 0.05$, two-way repeated measures ANOVA followed by holm-šídák's multiple comparisons test. The small circles denote the number of replicates; the numbers below the bars denote total number of animals. I. Number of eggs laid per female in 48 h. The small circles denote the number of eggs laid by an individual female. Data are shown as mean ± s.e.m. Student's *t*-test. Ns (not significant), $P > 0.05$.

We synthesized four mature MIP peptides for testing a possible role in the PMR in *N. lugens*. For this test, we injected a mix of MIPs (MIP1 – MIP4) at different doses into the abdominal hemocoel of virgin females and allowed them to recover for 6 hr in groups. This extended recovery time was required because virgins tested shortly after injection did not mate regardless of the substance injected. After recovery, injected females were exposed to wild-type males for 30 min, an exposure time that was sufficient for nearly all control females to show receptivity. Similar to results in *Drosophila* [17,18], we found that injecting a mix of MIPs into the abdominal hemocoel of virgin females does not decrease their receptivity (Fig 5C) or induce oviposition (Fig 5D and 5E).

### Silencing the MIP receptor gene does not affect the PMR of *N. lugens*

The MIP receptor (MIPR) has been reported to be involved in the PMR of *Drosophila* [16], tobacco cutworm [25] and cotton bollworm [24]. However, a recent study indicated that MIPR is not required for refractoriness to remating or induction of egg laying in *Aedes aegypti* [19]. The MIPR has been identified in most of insect species, including *N. lugens* [17,18,64] (S11 Fig) and the MIPR of *N. lugens* is orthologous to the *Drosophila* MIPR (S11 Fig). Hence, we asked whether the MIPR mediates the post-mating switch in the brown planthopper behavior. As a first step to address this question, we cloned the *mipr* gene of *N. lugens* (S12 Fig). The MIPR of the brown planthopper displays a typical seven transmembrane domain and it clusters with MIPR of other insect species (S12A and S12B Fig). We furthermore investigated the expression pattern of *mipr* in tissues of the brown planthopper. The *mipr* is found throughout the nymphal and adult stages, and is expressed more predominantly in the head than other tissues (S13A and S13B Fig).

Next, we tested whether the MIPR plays a role in the PMR of *N. lugens* using RNAi technique. The efficacy of the RNAi was assayed by qPCR of whole animals and we found that the expression of *mipr* was significantly reduced (Fig 5G). We used a protocol in which individual virgin females (*dsmipr* injected, or controls that were *dsgfp*- -injected) were first tested for receptivity towards a naive male (Fig 5F). Those females that mated were then allowed to lay eggs for 48 h before being retested for receptivity with a second naive male (Fig 5F). In the initial mating assays, virgin *mipr* RNAi females were as receptive as the control females (Fig 5H). When testing mated females for a second mating we did not detect any difference between controls and *N. lugens* with *mipr* knockdown, suggesting that the MIPR has no effect on the refractoriness to remating (Fig 5H).

Next, we investigated whether increased post-mating oviposition requires MIPR signaling by applying *mipr* RNAi. We found that both *dsgfp*- and *dsmipr*-injected females laid very few eggs if they were not mated (Fig 5I). We hypothesized that if the MIPR is required for post mating oviposition, *dsmipr*-injected females would lay few or no eggs even after mating. However, the number of eggs laid by mated *dsmipr*-injected females was not significantly different from that laid by

the controls (Fig 5I). We thus conclude that MIPs and MIPR play no obvious roles in the tested post mating response of *N. lugens*, and the MIPR is not likely to be the receptor of the novel peptide macc.

## Discussion

In insects, mating commonly induces a switch in the behavior and physiology of the female, referred to as a PMR. This PMR ensures a numerous and viable offspring and secures paternity for the male. However, only in fruit flies, the underlying mechanisms have been elucidated in detail, and the MAG-derived peptide sex peptide has been identified as a primary secretory signal [3,4,6,7,9,11–13]. Another closely related MAG-derived peptide, DUP99B, is also contributing to the PMR in *D. melanogaster* [69,70]. Since sex peptide and DUP99B can be found only in a few *Drosophila* species, we set out to identify factors in the MAG of the brown planthopper that might play a role similar to sex peptide/DUP99B in inducing a PMR First, we established that brown planthopper females display a distinct change in behavior and physiology after mating. This is manifested in a reduction in receptivity to mating males and an increase in ovulation lasting four days. Next, we showed that extract from MAG injected into female *N. lugens* induces a significant decrease in receptivity, lasting about 24 h, and a significant increase oviposition. Then we asked what the active factor in MAG extract that induces the PMR might be. We therefore performed a transcriptional analysis of MAG extract. As expected, sex peptide and MIPs were not detected in the MAG, but one splice form of ion transport peptide, ITPL, and a novel 51 amino-acid peptide were identified. A fragment of the latter was also detected by mass spectrometry. While the ITPL was identified also in other tissues of both males and females (see also [36,40]), we focused primarily on the novel peptide, maccessin, which is male-specific and only found in the MAG.

Virgin female *N. lugens* mated to males with *macc* knockdown do not display a repression of the propensity to re-mate, whereas injections of recombinant macc peptide into virgin females render them less receptive to courting males. However, oviposition was not affected by these manipulations. Thus, we propose that *macc* mediates a male signal transferred in seminal fluid to reduce female receptivity to further courting males. In our experiments this signal only mediates the receptivity part of the total PMR seen after regular mating and the duration of the effect is shorter (only 24 h instead of 4 d). We speculate that the partial and shorter PMR effect seen in our macc experiments could be for the following reasons: (1) in *Drosophila* sex peptide to exert its full effect over about a week needs to be bound to sperm when transferred to the female and thereafter gradually released [58], and thus macc injection without sperm may be less efficient and the peptide exposed to degradation by protease activity, (2) it is likely that macc is not the only MAG-secreted factor required for a full PMR effect and therefore *macc* RNAi in males is not sufficient. In fact, we did identify another peptide, ITPL-1 in the MAG of *N. lugens* and found that it also induces a weak and partial effect on the female PMR.

Since the receptor of sex peptide in *Drosophila* [16] can be activated also by MIPs [8,17,18], and the MIPR was implicated in the PMR in some insects [24,25], we tested whether sex peptide, MIPs or their receptor (MIPR) affects the PMR in *N. lugens*. We found no effect of manipulating MIP signaling or injections of *Drosophila* sex peptide and planthopper MIPs. The outcome of the MIPR knockdown experiment also suggests that this receptor is not required for macc signaling. Thus, since macc is a novel peptide with no sequence relation to previously identified peptides, its receptor is still unknown, and we were unable to manipulate this part of the signal pathway in female tissues for further tests.

It is intriguing that sex peptide and Dup99B are found only in the genomes of a few *Drosophila* species related to *D. melanogaster* [15,17,71]. Another seminal fluid protein that is also specific to *D. melanogaster*, ovulin, is sequentially cleaved into four smaller peptides [72,73], and stimulates egg laying in females the first 24 h after mating [74–77]. MAG-derived peptides regulating a PMR response have not yet been unequivocally identified in any other insect, except the mosquito *Aedes aegypti* where post-mating receptivity to males was found to be affected by a MAG-derived peptide, head peptide-1 [27], as detailed below. Since sex peptide acts on the MIPR several studies investigated the involvement of MIP signaling in a PMR in various insects. MIP activated MIPR signaling seems not to be a primary regulator of PMR in the insects studied (including the present study), although the MIPR and some unidentified ligand affects oviposition

and ovary development in some insects [19–22,25]. There is however a report on MIP expressing interneurons in abdominal ganglia of female *Drosophila* which promote receptivity to mating both in virgin and mated females [67]. In mosquitos a post-mating decline in female receptivity to further mating attempts is mediated by the MAG-derived peptide head peptide-1 (a form of short neuropeptide F) and the short neuropeptide F receptor [27]. However, the head peptide-1 signaling does not affect fecundity, host-seeking or blood-feeding. This is similar to the brown planthopper where *macc* is MAG-derived signal that reduces female receptivity, but not fecundity. Thus, mosquito head peptide-1 and brown planthopper macc may be partial functional analogs of sex peptide.

What sex peptide, DUP99B, ovulin, head peptide-1 and macc have in common is that they are MAG-derived peptides that appear to be restricted phylogenetically. We found a macc precursor transcript only in the genomes of *Nilaparvata lugens* and *Nilaparvata muiri,* but not in the related small brown rice planthopper, *Laodelphax striatellus*, or the white-backed planthopper, *Sogatella furcifera,* or other more distantly related insects. Apparently MAG-derived secretory peptides undergo rapid evolution in certain species and in *Drosophila* sex peptide repurposes an already existing receptor (MIPR) for distantly related MIP neuropeptides [15,17,18,71]. Similarly, *Aedes* head peptide-1, has adopted a short neuropeptide F receptor [27]. Thus, sex peptide displays some sequence similarities to MIPs [17,18] and head peptide-1 is short neuropeptide F-like, while the macc sequence is unique making it a more complex task to select known GPCRs (or orphan receptors) for screening. The role of ITPL-1 needs to be further investigated. Since it also seems to be produced in female *N. lugens* from the other splice variants itpl-2 – 4, it may act both via transfer from males at copulation and as an endogenously secreted peptide in females. This would resemble the situation in *Drosophila* where endogenous MIP produced in female-specific neurons also regulates mating receptivity [67] (indicating a dual role of MIP receptors as targets of both sex peptide and MIP in reproductive behavior). Receptors for ITPa and ITPL peptides have been identified in the moth *Bombyx mori* and the fly *Drosophila* [78,79]. Thus, receptor knockdown could be attempted in females for tests of a role in the PMR in the future.

In summary, we identified a novel peptide, macc, in the MAG of *N. lugens* that induces a PMR in mated females rendering them less receptive to further mating attempts. It remains to identify a receptor for this novel peptide and to characterize target circuits in the central nervous system that modulate the female behavior. Additionally, the role of ITPL-1 in the PMR should be further investigated, including the possible role of endogenous female ITPL-1 in regulating reproductive behavior and physiology. Furthermore, we need to search for additional factors that ensures an increased fecundity in mated females and leads to a more complete PMR resembling that seen after mating.

## Materials and methods

### Experimental insects and husbandry

The brown planthopper, *N. lugens,* was reared on 'Taichung Native 1' (TN1) rice (*Oryza sativa* L.) seedlings in our laboratory and maintained at 27 ± 1 ℃, with 70 ± 10% relative humidity, under a 16 h: 8 h light dark photoperiod [80]. Both virgin females and males after adult eclosion were collected for receptivity and fecundity analysis.

### Collection of male accessory glands

Adult virgin males from 3 to 5 days after emergence were dissected in phosphate buffer (pH 7.2) under a stereomicroscope (Leica S8AP0). The accessory glands (Fig 1C) were removed with forceps into Eppendorf tube followed by an addition of 500 µL 80% methyl alcohol and a 5-mm stainless steel beads (Qiagen, Germany). The mixture was homogenized 2 min using a TissueLyser (Qiagen, Germany) followed by 10-min intermittent sonication in an ice bath. The supernatant was collected after centrifugation for 5 min (10,000 g, 4°C). This extraction procedure was further repeated twice and these supernatants from three extractions were combined as one sample. The sample was concentrated in a SpeedVac vacuum concentrator, and dried in a SpeedVac. The dried samples were stored at −80°C until use. The purified male accessory glands extracts were taken up in phosphate buffer to a final concentration of 10 accessary gland equivalents per microliter.

### RNA-seq and data analysis

Total RNA was extracted from male accessory glands of each unmated brown planthopper male from 3 to 5 days after emergence, with each sample consisting of 150 male accessory glands as a biological replicate. Three biological replicates were prepared. RNA extraction was performed using the TRIzol reagent (Invitrogen) following the manufacturer's protocol. RNA quality and quantity were assessed using an Agilent 2100 BioAnalyzer (Agilent, USA), and RNA integrity was confirmed through electrophoresis on a 1% agarose gel. RNA-seq libraries construction, sequencing and assembly of transcriptome reads was performed by Novogene with Illumina HiSeq2000 platform (Novogene Bioinformatics Technology Co.Ltd, Beijing, China) and Trinity (v2.4.0), respectively.

The clean reads were generated by removing adapters, poly-N, and low-quality reads from the raw data using fastp algorithm (v0.12.4). They were then mapped to the reference genome of *N. lugens* using Bowtie2 [37]. The annotation of unigenes were performed as described previously [81]. The gene expression levels were normalized to FPKM (Fragments Per Kilobase of transcript sequence per Millions base pairs sequenced) using DESeq2 R package [82]. The Gene Ontology (GO) enrichment analysis and KEGG pathway analysis were performed using GOSeq (v1.10.0) and KOBAS (v2.0.12).

### Proteomic analysis

Proteomic sequencing was performed using Label-free LC-MS/MS method [61] based on platform of Novogene Co., Ltd. The detailed procedure was described below.

### Male accessory glands protein extraction

Three hundred MAGs were dissected using the same method as mentioned above. Frozen MAGs were ground in liquid nitrogen before the addition of 500 μL protein cracking liquid (100 mmol/L $NH_4HCO_3$, 8 mmol/L urea, 0.2% SDS, pH 8.0). The protein extracts were centrifuged at 12,000 g for 15 min at 4°C and the supernatants were collected for the analysis. Protein concentration was determined with a Bradford Protein Colorimetric Assay Kit using BSA as a standard and a wavelength of 595 nm. Protein quality was determined with SDS-PAGE. Supernatants were stored at − 80 °C until use.

### In-solution protein tryptic digestion

A protein sample was suspended in 10 mmol/L DTT and boiled at 56°C for 1 hour. The soluble samples were alkylated with 200 mM IAA for 1 h. Digestion was performed using the 2.5 μg trypsin in $NH_4HCO_3$ buffer overnight at 37 °C. The digested peptides were collected by centrifugation at 12,000 × g for 5 min. The digested peptides were desalted using C18 pipette tip (Thermo Scientific) according to manufacturer's protocol. Then, the samples were concentrated by vacuum drying and dissolve in 0.1% (v/v) formic acid.

### LC–MS/MS analysis

LC–MS/MS analysis was performed as follow: The digested peptides (20 μL) were loaded onto the trap column at a flow rate of 10 μL/min by EASY-nLC 1200 system (Thermo Fisher Scientific). After trap equilibration, the samples were eluted with a linear gradient of buffer A (0.1% formic acid) and buffer B (80% acetonitrile and 0.1% formic acid) at a flow rate of 600 nL/min. Separated peptides were examined in the Q Exactive HF-X mass spectrometer (Thermo Scientific) with a Nanospray Flex ionization source. Spectra were scanned over the m/z range 350–1500 Da at 60,000 resolution. 20 s exclusion time and 27% normalization collision energy were set at the dynamic exclusion window.

RAW files were extracted using the MASCOT version 2.3.02 (Matrix Science, London, UK) embedded into Proteome Discover 2.2 (PD2.2, Thermo Scientific). MS data were searched against a *N. lugens* MAG transcriptome database. The search parameters were set as follow: fully tryptic peptides with ≤ 2 missed cleavages were permitted; carbamidomethylation (C) and oxidization (M) were as fixed and variable modifications, respectively; peptide mass tolerance was 10 ppm,

and fragment mass tolerance was 0.02 Da and carbamidomethyl as a static modification. Peptide Spectrum Matches (PSMs) with a confidence of more than 99% were classified as trusted PSMs. The cutoff of global false discovery rate (FDR) for peptide and protein identification was set to 0.01. Subsequently, an association analysis with the transcriptome data was conducted to elucidate the relationships between protein expression and gene sequences.

## Bioinformatic analysis

The SignalP 4.1 Server was used to predict potential signal peptides (http://www.cbs.dtu.dk/services/SignalP/). Interproscan software [63] was used for GO and IPR functional annotation (including Pfam, PRINTS, ProDom, SMART, ProSite, PANTHER databases). Protein pathway analysis was performed using the COG database (https://www.ncbi.nlm.nih.gov/research/cog/) and Kyoto Encyclopedia of Genes and Genomes KEGG (http://www.genome.jp/kegg). We used STRING DB software to predict possible protein-protein interactions (http://STRING.embl.de/) [83].

## Gene cloning and sequence analysis

We used the NCBI database and BLAST programs for sequence alignment and analysis. Then we used EditSeq to predict Open Reading Frames (orfs). The primers were designed by tools in NCBI. The primer sequence information is described in S3 Table. According to the manufacturer's instructions, total RNA was extracted by TRIzol reagents (Invitrogen, Carlsbad, CA, USA). We used HiScript III RT SuperMix for qPCR (+gDNA wiper) (Vazyme, Nanjing, China) reverse transcription kit to synthesize cDNA templates for cloning, and stored the synthesized cDNA templates at -20°C.

We predicted protein transmembrane fragments and topological structures through TMHMM v2.0 (http://www.cbs.dtu.dk/services/TMHMM-2.0/) [84,85]. Multiple alignments on the complete amino acid sequences were performed using ClustalX (http://www.clustal.org/clustal2/). The phylogenetic tree was constructed using MEGA 10.0 software and the Maximum Likelihood Method, with 1000 repeated starts.

## Gene expression profile analysis

For the stage-specific expression study of *mip* and *mipr*, total RNA were extracted from pools of thirty individuals from the following developmental stages: 1st to 5th instar nymphs, adult male and female insects. For the tissue-specific expression study of *mip* and *mipr*, total RNA was isolated from various tissues including head, thorax and abdomen of three-day-old male adults, and head, thorax and abdomen of three-day-old virgin female adults. For conducting a tissue-specific expression analysis of *maccessin*, total RNA was extracted from pooled samples of multiple individuals across the following phases: virgin females and males three days post-emergence, male accessory glands, and male bodies with removed accessory glands. All samples were extracted by using TRIzol reagent (Invitrogen).

## Quantitative RT-PCR

The first-strand cDNA was synthesized with HiScript II Q RT SuperMix for qPCR (+gDNA wiper) kit (Vazyme, Nanjing, China) using an oligo(dT)18 primer and 500 ng total RNA template in a 10 µl reaction, following the manufacturer's instructions. Real-time qPCRs of the various samples used the UltraSYBR Mixture (with ROX) Kit (CWBIO, Beijing, China). The PCR was performed in 20 µl mixture including 4 µl of 10-fold diluted cDNA, 1µl of each primer (10 µM), 10 µl 2×UltraSYBR Mixture, and 6 µl RNase-free water. The PCR conditions used were as follows: initial incubation at 95°C for 10 min, followed by 40 cycles of 95°C for 10 s and 60°C for 45 s. *N. lugens* 18S rRNA was used as an internal control. Relative quantification was performed via the comparative $2^{-\triangle\triangle CT}$ method [85].

## RNA interference

For lab-synthesized dsRNA, *gfp, mipr, itp, itp/itpl* and *Maccessin* fragments were amplified by PCR using specific primers conjugated with the T7 RNA polymerase promoter (primers listed in S3 Table). The dsRNA was

synthesized by a kit (MEGAscript T7 transcription kit, Ambion) according to the manufacturer's instructions. The integrity and quantity of the double-stranded RNA (dsRNA) products were confirmed using 1% agarose gel electrophoresis and a Nanodrop 1000 spectrophotometer. Subsequently, the samples were stored at -70°C until further use.

In order to silence target genes, 5 µg/µl dsRNA was injected into brown planthopper using a Micro4 Controller (WPI) with a Micro Pump, male 40nl, female 50nl, and control group the same amount of *dsgfp*.

## Brown planthopper injections

Fifth instar nymphs (3 days old) of the selected male *N. lugens* were anesthetized with carbon dioxide and injected into the tarsus under a stereo microscope with an injection system. The injected *N. lugens* were gently transferred to a culture bottle and placed in a constant temperature incubator at 26°C.

Total RNA was individually collected from each injected insect after reception assay, followed by extraction. The efficiency of gene silencing was subsequently assessed through qPCR.

## Peptide synthesis

Peptides were synthesized by Genscript (Nanjing, China) Co., Ltd. Myoinhibitory Peptides and Sex Peptide mass was confirmed by MS and the amount of peptide was quantified by amino acid analysis. Ion transport peptides and maccessin peptide were generated as recombinant proteins expressed in CHO cells. These proteins were purified by AmMag Ni Magnetic Beads. The amino acid sequence of the peptides used in this study are: *N. lugens* Myoinhibitory Peptide 1: (MIP1): AWRDLQSSWamide; Myoinhibitory Peptide 2: (MIP2): GWQDMPSSGWamide; Myoinhibitory Peptide 3 (MIP3): GWQDLQGGWamide; Myoinhibitory Peptide 4 (MIP2): AWSSLRGTWamide; *D. melanogaster* Sex Peptide: (SP): WEWPWNRK{Hyp}TKF{Hyp}I{Hyp}S{Hyp}N{Hyp}RDKWCRLNLGPAWGGRC. Mature proteins ITPa (comprising amino acids 23–113), ITPL (comprising amino acids 23–117), and maccessin (comprising amino acids 20–91), each fused with a 6xHis tag at their C-termini (designated as protein-His tag), were expressed in Chinese Hamster Ovary (CHO) cells. These proteins were subsequently purified using AmMag Ni Magnetic Beads. Additionally, the mature ITPa protein with an amide modification at the C-terminus (referred to as ITPa-amide) was also generated in CHO cells. All these proteins were codon-optimized for expression in mammalian cells.

## Behaviour assays

The behavioral experiments were all conducted in a greenhouse at 26°C. Following the daily activity of the brown planthopper, the mating experiments were scheduled between 3 p.m. to 7 p.m.

**1. Post mating response.** The 5th instar nymphs were selected from the feeding box and fed on 5–6 cm high rice seedlings, and the emerging males and females were collected every 12 hours for single-sex feeding. The reproductive system of the emerging adult is not fully developed, and studies have shown that the mating rate of the adult brown planthopper 24 hours after emergence is less than 10% [31]. These early-emerged females were utilized as virgin test subjects for our experiments.

In the initial mating experiment, a pair of unmated male and female individuals (3 days after emergence) were placed in a 24 mm diameter x 95 mm high transparent fruit fly tube, shared with rice seedlings for 30 minutes. The mating process was observed during this time and the mating success rate was recorded and calculated. Post-mating females and an unmated male of the same age were transferred to new transparent tubes containing rice seedlings 24 hours after initial mating, while unmated virgin females of the same age served as controls. We observed the mating process and record the mating success rate. The re-mating experiment was conducted every 24 hours, and the same procedure was followed for 5 days. The mating process was observed and videotaped to record the mating success rate of each group.

For the egg-laying experiments, unmated or mated females (3 days after emergence) were placed in a 24 mm x 95 mm clear *Drosophila* tubes with rice seedlings. Each female was numbered and transferred to a new *Drosophila* tube with rice seedlings every 24 hours. The number of eggs in the rice seedlings were counted.

**2. Tests if MAG extract induces a post-mating response.** For experiments to test MAG extract-injection, virgin females that had eclosed 3 days earlier were selected for injection. Receptivity assays were performed with males of the same age placed in single pairs in the tube 3h, 6h, 12h, 24h and 36h after injection. Each female was injected with the equivalent of half of accessory gland. On the day after eclosion, each virgin was injected with 30 nl MAG extract or solvent and placed in the transparent circular tube with rice seedling to lay eggs for 24 h.

**3. Peptide injection to test virgin receptivity.** For experiments using peptide-injected females (3 days after eclosion), the mating experiment was conducted 6 hours after injection, when females had fully recovered from the wound.

On the day after eclosion, each virgin female was administered an intra-abdominal injection of 30 nl of mature peptide or PBS. Subsequently, they were housed in a transparent cylindrical tube with a rice seedling, where they were allowed to oviposit for a period of 48 hours.

**4. Effect of silencing female *mipr* gene on post-mating response.** For the effect of silencing the female brown planthopper *mipr* gene on the post-mating response, the experimental protocol is shown in Fig 2D. After recovering for 1 day, the virgin females were mated with wild type males. The mated females were re-mated with unmated males 2 days after the first receptivity assay. Between the first mating and the re-mating assay, mated females were placed in tubes with rice seedlings to lay eggs. Samples were collected within 2 hours after re-mating assay and testing the interference efficiency. The expression level thus reflects the expression level of *mipr* during the experiment.

**5. Effect of silencing the male *maccessin* gene on post-mating response.** As for experiments using dsRNA-injected males, co-caging with female virgins of the same age was for 3–4 days after injection, when the *maccessin* gene was silenced to the greatest extent. Males who had eclosed within 24 hours after dsRNA injection (meaning they were at the end of their fifth age at the time of injection) were selected for feeding and mating with virgin females two days after eclosion. Samples were collected immediately after mating for gene expression measurement. Females that successfully mated were collected and fed individually. The number of eggs laid by females 48 hours after successful mating were counted. Mated females were remated with wild-type virgin males 2 days later.

## Statistics

We employed GraphPad Prism 9 software for data visualization and statistical analysis. Data presented in this study were first verified for normal distribution by D'Agostino– Pearson normality test. If normally distributed, Student's *t* test was used for pairwise comparisons, and one-way ANOVA or two-way ANOVA was used for comparisons among multiple groups, followed by Tukey's multiple comparisons. If not normally distributed, Mann–Whitney test was used for pairwise comparisons, and Kruskal–Wallis test was used for comparisons among multiple groups, followed by Dunn's multiple comparisons. All data are presented as mean ± s.e.m. All data are collected from at least four independent experiments. Every independent experiment used at least ten insects.

## Supporting information

**S1 Video. Courtship behavior of brown planthopper.**
(MP4)

**S2 Video. Post-mating behavior of brown planthopper.**
(MP4)

**S3 Video. Post-mating behavior of virgin brown planthopper after injection with MAG extract.**
(MP4)

**S1 Table. Summary of sequence assembly after RNA-seq of male accessory glands.**
(XLSX)

**S2 Table. Identified Seminal fluid proteins of *N. lugens.***
(XLSX)

**S3 Table. Primer sequences used in this study.**
(XLSX)

**S4 Table. Liquid chromatographic elution gradient table.**
(XLSX)

**S5 Table. Raw data for figure production of this paper.**
(XLSX)

**S1 Fig. Mating and post-mating behavior of brown planthoppers. A-H**: The mating behavior sequence. The sequence includes eight steps (A-H, following, wing extension, abdominal vibration, abdominal rubbing, attempted copulation, copulation, terminated copulation and leaving). The larger individual is the female and the smaller is the male. **I** and **J**: The post-mating response behaviors.
(TIF)

**S2 Fig. Labeling of sperm in the female copulatory bursa.** The sperm were stained with DAPI (red circle in the mated female). Scale bar: 100 μm.
(TIF)

**S3 Fig. (A) Workflow for identification and quantitation of seminal fluid proteins in the *N. lugens*.** Dissected accessory glands were used for extraction and subjected to transcriptome and proteome analysis using liquid chromatography (LC) and mass spectrometry (MS/MS). (B) Venn diagram of the numbers of predicted seminal fluid proteins comparing transcriptome prediction and MS identification.
(TIF)

**S4 Fig. Pathway assignment based on KEGG analysis.** (A) Classification based on transcript. (B) Classification based on protein.
(TIF)

**S5 Fig. Receptivity of virgin and mated females after seminal fluid protein candidate genes were knocked down by dsRNA injections, scored as the percentage of females that copulated within 30 min. The different genes are referred to by their numbers.** The small circles denote the number of replicates; the numbers below the bars denote total number of animals. Data are shown as mean ± s.e.m. *$P < 0.01$, and ns (non-significant), $P > 0.05$, two-way repeated measures ANOVA followed by šídák's multiple comparisons test. The annotations of these seminal fluid genes were showed below.Sfp33906: seminal fluid protein (APA33906.1). Sfp45778: melanization protease 1-like. Sfp48992: N(3)-methylcytidine methyltransferase METTL6. Sfp21953: GTP-binding nuclear protein Ran. Sfp21797: seminal fluid protein (APA33927.1). Sfp23008: seminal fluid protein (APA33928.1).
(TIF)

**S6 Fig. The presence of maccessin in different species.** (A) The presence of maccessin in different species. (B) The alignment of maccessin protein sequence between *Nilaparvata muiri* (upper) and *Nilaparvata lugens* (lower).
(TIF)

**S7 Fig. Sequence analysis of ITP/ITPL. A** The identified ion transport peptide/ ITP-like (ITPa/ITPL) transcripts in the brown planthopper. The coloured blocks represent exons within the ITP/ITPL transcripts. Exons 1a, 1b, 1c and 1d are

alternative 5' untranslated regions used by ITP and ITPLs. * denote the start codon and stop codons of the transcripts. **B.** Amino acid sequence of ITP and ITPL of the brown planthopper. Orange indicates sequence of signal peptide; green indicates mature peptide sequence; red indicates difference sequence of ITP and ITPL. Note that four slice forms of *itpl* are known (*itpl-1–4*), which all could give rise to the same mature ITPL peptide. **C.** Multiple comparisons of ITP and ITPL mature peptides in brown planthopper and other species. The red frames indicate conserved cysteines. Deduced ITP and ITPL sequences are shown for *Manduca sexta* (Manse, AY950500, AY950501), *Bombyx mori* (Bommo, AY950502, AY950503), *Schistocerca gregaria* (Schgr, XP_049859893.1), *Apis mellifera* (Apime, XP_006571870.1), *Aedes aegypti* (Aedae, AY950504, AY950505, AY950506), *Anopheles gambiae* (Anoga, EAA09451.4, XP_061497799.1), *Drosophila melanogaster* (Drome, NP_001036569.2, ABZ88141.1, NP_001163293.1), and *Tribolium castaneum* (Trica, EFA07585). (TIF)

**S8 Fig. Distribution of *itpl-1* transcript in different sexes and tissues of *N. lugens*. A.** Relative expression of *itpl-1* gene in *N. lugens* at different sexes. Data are shown as mean±s.e.m. Student's *t*-test. ****, $P<0.0001$. **B.** Relative expression of *itpl-1* gene in *N. lugens* in different tissues in the male reproductive system. Data are shown as means±s.e.m. Groups that share at least one letter are statistically indistinguishable; Kruskal–Wallis test followed by Dunn's multiple comparisons test with $P<0.05$. **C.** The tissue distribution of *itpl-1* analyzed by semi-quantitative RT-PCR. RNA samples from adult females, adult males, male accessary gland (AG) alone and adult male without accessary gland (male - AG). **D-G.** Relative expression of different spliceosomes of *itpl-1–4* gene in males injected with dsRNA. Data are shown as mean±s.e.m. Groups that share at least one letter are statistically indistinguishable; Kruskal–Wallis test followed by Dunn's multiple comparisons test with $P<0.05$. (TIF)

**S9 Fig. Nucleotide and amino acid sequence of the brown planthopper *mip* precursor gene (*mip*).** The four distinct mature peptides—MIP1 (green), MIP2 (yellow), MIP3 (purple), and MIP4 (cyan) are distinguished by unique colors. Cleavage sites (KR) are denoted by rectangular boxes, while the glycine residues (G) essential for amidation are highlighted by double underlining. (TIF)

**S10 Fig. Relative quantification of *mip* transcript levels in different developmental stages (A) and tissues (B) of brown planthopper. A.** Relative expression of *Nlmip* gene in brown planthopper at different developmental stages. **B.** Relative expression of *Nlmip* gene in brown planthopper in different tissues. In the analysis of spatiotemporal expression patterns of *mip*, there were no less than 10 brown planthopper in each sample. M: male; F: female. Data are shown as means±s.e.m. The presence of the same letter on the column indicates no significant difference between the two groups, and the absence of the same letter indicates a significant difference between the two groups. (TIF)

**S11 Fig. The presence of sex peptide receptor/MIPR and MIP in different species.** The brown planthopper (marked in red) lacks sex peptide. In Hymenopteran insects (marked in green), the honey bee *Apis mellifera* and the parasitic wasp *Nasonia vitripennis*, neither sex peptide, MIP, nor sex peptide receptor/MIPR are found. The "+" symbol represents presence, while the "-" symbol indicates absence. The phylogenetic tree of different insect species has been made by NCBI CommonTree https://www.ncbi.nlm.nih.gov/Taxonomy/CommonTree/wwwcmt.cgi. (TIF)

**S12 Fig. Sequence analysis of MIPRs. A** Multiple comparison of MIPR in brown planthopper (NLMIPR) and four other insect species (*Bombyx mori*, *Drosophila melanogaster*, *Amyelois transitella* and *Tribolium castaneum*). The black lines (TM1-TM7) depict the transmembrane domains. **B.** Phylogenetic analysis of MIPRs in different insect species. (TIF)

**S13 Fig. MIP receptor expression in *N. lugens*: relative quantification of *mipr* transcript levels in different developmental stages (A) and tissues (B) of *N. lugens*. A.** Relative expression of *mipr* gene in brown planthopper at different developmental stages. **B.** Relative expression of the *mipr* gene in different tissues of adult brown planthopper. M: male; F: female. Data are shown as means±s.e.m. The presence of the same letter on the column indicates no significant difference between the two groups, and the absence of the same letter indicates a significant difference between the two groups.
(TIF)

## Acknowledgments

We thank Dr. Mariana Wolfner for commenting on an earlier version of this paper.

## Author contributions

**Conceptualization:** Dick R. Nässel, Shun-Fan Wu.

**Data curation:** Yi-Jie Zhang, Ning Zhang, Ruo-Tong Bu, Shun-Fan Wu.

**Formal analysis:** Yi-Jie Zhang, Shun-Fan Wu.

**Funding acquisition:** Cong-Fen Gao, Shun-Fan Wu.

**Investigation:** Yi-Jie Zhang, Ning Zhang, Ruo-Tong Bu.

**Supervision:** Dick R. Nässel, Cong-Fen Gao, Shun-Fan Wu.

**Validation:** Yi-Jie Zhang, Ning Zhang, Ruo-Tong Bu.

**Writing – original draft:** Yi-Jie Zhang, Dick R. Nässel, Shun-Fan Wu.

**Writing – review & editing:** Dick R. Nässel, Shun-Fan Wu.

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
