## [Decision Letter · Decision Letter 0]

3 Dec 2024

PGENETICS-D-24-01238

A novel male accessory gland peptide reduces female post-mating receptivity in the brown planthopper

PLOS Genetics

Dear Dr. Wu,

Thank you for submitting your manuscript to PLOS Genetics. After careful consideration, we feel that it has merit but does not fully meet PLOS Genetics's publication criteria as it currently stands. Therefore, we invite you to submit a revised version of the manuscript that addresses the points raised during the review process.

Please submit your revised manuscript within 60 days Feb 01 2025 11:59PM. If you will need more time than this to complete your revisions, please reply to this message or contact the journal office at plosgenetics@plos.org. Please include the following items when submitting your revised manuscript:

We look forward to receiving your revised manuscript.

Kind regards,

Takaaki Daimon

Academic Editor

PLOS Genetics

Pablo Wappner

Section Editor

PLOS Genetics

Aimée Dudley

Editor-in-Chief

PLOS Genetics

Anne Goriely

Editor-in-Chief

PLOS Genetics

**Additional Editor Comments:**

The submitted manuscript has been evaluated by three reviewers.

As detailed below, all reviewers agreed that the manuscript presents significant advances in understanding post-mating responses in insects.

Therefore, I kindly request the authors to revise the manuscript accordingly.

Please note that Reviewer #2 has attached a file highlighting potential changes to the manuscript.

**Journal Requirements:**

1) We do not publish any copyright or trademark symbols that usually accompany proprietary names, eg ©,  ®, or TM  (e.g. next to drug or reagent names). Therefore please remove all instances of trademark/copyright symbols throughout the text, including:

- ® on Line: 872.

- TM on Lines: 807, and 810.

4) We notice that your supplementary Figures are included in the manuscript file. Please remove them and upload them with the file type 'Supporting Information'. Please ensure that each Supporting Information file has a legend listed in the manuscript after the references list.

Potential Copyright Issues:

i) Please confirm (a) that you are the photographer of Figures: 1C, 1D, Supp. 1 - Figures: 2 Supp. 1A - Figures 5A, and and 5B., or (b) provide written permission from the photographer to publish the photo(s) under our CC BY 4.0 license.

6) Please amend your detailed Financial Disclosure statement. This is published with the article. It must therefore be completed in full sentences and contain the exact wording you wish to be published. Please ensure that the funders and grant numbers match between the Financial Disclosure field and the Funding Information tab in your submission form. Note that the funders must be provided in the same order in both places as well.

- State the initials, alongside each funding source, of each author to receive each grant. For example: "This work was supported by the National Institutes of Health (####### to AM; ###### to CJ) and the National Science Foundation (###### to AM).".

**Reviewers' comments:**

Reviewer's Responses to Questions

**Comments to the Authors:**

Reviewer #1: The goal of this study was to characterize post mating responses in the brown planthopper, an important pest species, and determine the molecules involved in responses. Building on the existing body of knowledge regarding post-mating responses in Drosophila, they searched for a sex-peptide like molecule in the male brown planthopper seminal fluid that would induce post mating responses in females. They also characterized behavioral post mating stages in brown planthoppers and conducted a transcriptional and proteomic analysis of the male accessory gland fluid, followed by functional analysis of two candidate peptides (maccessin and ITPL-1 ). The authors ultimately determined that the newly discovered maccessin and ITPL-1 reduce female mating receptivity. No effects on the other post mating response tested, oviposition, were observed. The authors convincingly demonstrated effects of these peptides through transient knockdown with RNAi as well as injection of synthetic peptides. The authors then went on to investigate a receptor for maccessin, starting with an evaluation of myoinhibitory peptide receptors. They cloned the mip gene from the brown planthopper and characterized expression patterns and then injected the synthesized MIP peptides into females to no effect. Overall, this paper represents a vast source of new information on post mating responses in the brown plant hopper and should be of interest to researchers working on insect post mating responses. The manuscript could be improved in several ways including addition of significant methodological details, citations and statistical results.

Introduction:

LN 64. add “and others” to reflect the additional PMR phenotypes not listed here

LN 94. This statement is not accurate, the study by Duvall and others demonstrated that HP-1 has a short effect (less than 24hrs) on enforcing paternity. Other unknown factors control long-term refractory behavior.

Results:

LN 162. The resistance to courting wears off after 24 hours in MAG-injected females. However, did you check all the time points in Fig.1A to be sure that there is no long-term effect of MAG injection?

Fig.1. mean circles are hard to see (1B), what do the bars represent? I think SEM based on line 994, but this should be stated in the figure caption.

170: Figure 1: more detail is needed on how categorization of females as either mated or virgin was. Confirmed. Did you check for presence of sperm in the spermatheca? Is courted a suitable proxy for mating?

Figure 1B: student’s t-test may not be the appropriate statistical test here since there are repeated measures.

LN 179-180. This statement needs a citation.

LN 188-189. “The number of refractory females is high 3-6 h after SFP 189 injection, but then declines.” How is this possible if in the methods (LN 344-345), the authors state that females need 6 hrs to recover from injections?

LN 195. Fig. 1. Given the high variation in egg laying is this sample size (only 5 insects) adequate? Was a power analysis performed to determine the appropriate sample size? Is it appropriate to combine replicates? Show the statistical result demonstrating no replicate effect and that combining all samples was appropriate.

LN 204. Indicate which is male and which is female (with an image, perhaps), for those less familiar with this insect.

LN 219-220: What program was used to perform these QC steps?

LN 223: Is there a reference you can cite for this genome?

LN 229: A reference needed here

LN 238: It is unclear what program was used for the GO terms analysis. More details are needed.

LN 273 Fig. 1. This figure is difficult to interpret with so many categories. Consider combining categories. Perhaps follow convention used by other authors reporting similar data. One option would be to group all the go terms with less than e.g. 5 % and term them “other”. Otherwise, the color key is not particularly useful as there are many similar hues. All raw data should be made available to readers in the supplement or elsewhere.

LN 276: Make the final sentence at the end of the figure legend more explanatory.

There are no statistics in this GO term analysis. Are any of the GO terms significantly overrepresented in the data?

LN290 Fig. S2. What are the categories “overview” and “overview maps”?

LN 302: In which tissue is the other variant of this gene produced in females?

LN 332: How many days post injection was the reduction in expression recorded? How long did the reduction in expression last?

LN 305-306. More information on why these peptides were considered top candidates for further evaluation would be helpful. Were other molecules evaluated, and the data are just not shown?

Figure 3A: The description of the different colors can go in the figure legend. It would be clearer just to show the amino acid sequence here.

Figure 3B: I assume the female used for RT-PCR was virgin, though it is unclear. This should be made clearer either in the figure or in the legend.

Fig 3C: what time point post dsRNA injection were samples collected for qPCR? Is it 2 days post eclosion as this is when the egg laying assay was done? This needs to be clarified.

Fig 3D. “Such re-mating is never observed in the control (dsgfp-injected) group. However, the re-mating rate is small with only 7 percent of the females courted in the secondary mating being receptive” is this number statistically significant?

Figure E/G: the markers on the graph denoting the mean are very hard to see. I don’t see that the standard error is shown on either graph, as suggested in the figure legend.

Figure 3 F/G/H: It would be clearer if the x axis labels were PBS-injected of Maccessin-injected. Otherwise, the difference between the dsRNA injections vs this new treatment is not clear.

Figure 3 H: if there were multiple replicates performed for this assay, you could show them on the graph, as is the case of other panels of the figure.

Figure 3: In general, the positioning of the different elements of this figure are misaligned. Adjust position of graphs E, F and G and H so the X axes are in line with one another.

LN 356. Move information from A into the legend as this output is not necessary. It would be clearer just to show the amino acid sequence here.

LN 423-427. Was this result statistically significant?

LN 445: Here, and elsewhere in the results section, statements are made about the data that are not accompanied by statistical tests. This needs to be addressed.

LN 503. Fig 4S. Gender is a social norm for humans. Sex is more appropriate here.

LN 535-539. In the expression analysis of mip at different life stages, it’s not clear in the text or in the graph how many biological replicates of this experiments were performed, nor how many individuals were used. There are no statistical tests performed here either to support the claims made that mip is more abundant in heads of males and females.

LN 559: Sometimes mipr is in caps/italics and sometimes not. This needs to be consistent.

LN 562-565: Satistics and/or sample sizes need to be included to support this statement.

LN 607-608. Levels of kd can vary over time. More details on when kd was checked for these insects is needed; was kd level confirmed at the same day and time as the experiment was conducted.

Figure 5G: please include in the figure legend or in the text at what point RNAi efficiency was measured. Was RNAi efficiency measured at the point of the receptivity assay and the remating assay? As I understand these assays were performed on different days. Do you know if the gene is still silenced at that point?

Figure 5G: same comment as above about the size of the mean/s.e symbols.

Figure 5 supp2: In panel A there are no statistical tests to compare expression between life stages. It would be clearer to display the individual biological replicates on the graph here too, as in B. This also applies to Figure 5 supp 5 A.

Discussion

LN 660. The opening sentence of the discussion is difficult to read. This should be split into two sentences.

LN 709. should read “was found to be affected”

Methods

LN 770. Define IAM

LN 755. The description of MAG protein extraction needs more detail; to replicate this study it would be helpful for a reader to know centrifugation speeds and times, for example.

LN 846: Primer sequences do not appear to be included in the supplemental data and need to be made available. This applies also to the dsRNA primers referenced on line 885

LN 996. Cite the reference for the Nilaparvata lugens genome

LN 973. A citation is needed for the DESeq2 R package

General comments and recommendations:

Present the statistical tests and results properly and consistently throughout.

More details on the methods are necessary in order to meet the rigor and reproducibility required by the journal.

Use a consistent designation of the study organism. In some places BPH is used and in other N. lugens is used.

Levels of kd can vary over time. More details on when kd was checked for these insects is needed; was kd level confirmed at the same day and time as the experiments were conducted?

Reviewer #2: Dear editor and authors,

I have completed the review of the article entitled "A novel male accessory gland peptide reduces female post-mating receptivity in the brown planthopper”.

The work is well written and the data package is complete and solid. The authors address a very interesting topic from a theoretical and practical point of view. On the one hand, they describe a new protein found in the accessory glands of the males of a pest species, which inhibits the receptivity of the females after copulation but they go further, trying to elucidate the receptors that protein in the female, with sophisticated proteomic and transcriptomic methods.

From a practical point of view, the work is in a rice pest species; work aimed at the study of seminal fluids transferred from the male to the females during copulation and its effect on the behavior and physiology of the females are key to the future development of environmentally friendly control strategies.

I attach the MS with minimal suggestions and I encourage the authors to continue along this line of research aimed to elucidate the specific receptors in females on which the molecules that inhibit remating and trigger oviposition after copulation act, since these are topics studied almost exclusively in Drosophila and a bunch of insect species.

Reviewer #3: The manuscript by Zhang et al characterizes the female post-mating response (PMR) of Nilaparvata lugens in terms of female sexual receptivity and oviposition, as well as identifies a seminal fluid protein, macc, responsible for decreasing female sexual receptivity. The authors conducted an impressive amount of work, including RNAi-mediated gene silencing and injections of synthetic proteins into virgin females, to address the role of macc in modulating the PMR. They further examined the effects of ITPL-1 (another seminal fluid protein of N. lugens), the Sex Peptide of Drosophila melanogaster and the myoinhibitory peptide allatostatin-B (MIP) along with its receptor (MIPR).

The results are potentially intriguing for readers of PLOS Genetics as they provide new insights into molecular insect biology, reproductive physiology, and evolutionary genetics, fields of broad interest. However, the manuscript is significantly hindered by poor text organization, inadequate background information, insufficient justification of certain analyses, and numerous syntax and punctuation issues. In its current form, the writing is unacceptable for publication. Any revision should be thoroughly edited by someone fluent in English. Therefore, I will not list minor errors or issues but outline below some comments and suggestions for improvement.

Introduction

1) Zhang et al. did not cite relevant studies that previously described the PMR in N. lugens. According to Yu et al. (2016; https://doi.org/10.1186/s12864-016-3013-7), mated females of this species exhibit stimulated oviposition levels (Long et al., 2010) and prolonged refractoriness to further insemination (Ichikawa, 1979). Although I could not access these articles, these findings indicate that the PMR has been studied before. The authors should incorporate this literature into their introduction to provide proper context and avoid presenting their results as if the PMR in N. lugens were previously uncharacterized.

2) The repertoire of seminal fluid proteins of N. lugens was already explored using transcriptomic and proteomic approaches (Yu et al., 2016). The authors should clarify why they conducted a new screen for male accessory gland (MAG) seminal fluid proteins.

3) Zhang et al. assert that macc is "the only candidate seminal fluid peptide that promotes a PMR in N. lugens." However, they also report similar effects for ITPL-1 on female receptivity. Since both proteins influence the PMR, it is incorrect to claim exclusivity for macc. Furthermore, other MAG-derived proteins may also play a role in the PMR. For instance, Ge et al. (2019) identified a MAG selenoprotein that affects oviposition levels in N. lugens. This relevant study is missing from the manuscript.

4) The last paragraph of the introduction is overly focused on describing results and is therefore misplaced. It should be rewritten or moved to the Results section.

5) The rationale for selecting macc as the focal protein is unclear. Was macc chosen because it was a novel seminal fluid protein, or were there other reasons? This should be explicitly explained to help readers understand the experimental framework. Additionally, the manuscript disproportionately emphasizes macc while treating ITPL-1 as secondary, even though both were studied using similar approaches and yielded comparable effects. The authors may consider separating the ITPL-1 findings into a distinct manuscript.

Results

6) The response variable used for characterizing female receptivity is unclear. Sometimes, the authors refer to the female willingness to accept courting males, giving the impression that they measured the female propensity to (re)mate, which is a reasonable proxy for receptivity. However, Fig. 1A seems to show the "Proportion of females courted", which is more related to attractiveness than to receptivity. Moreover, the Methods section vaguely describes the post-mating response assay: "The mated females were subjected to re-mating assay every 24 hours for 1–5 days after first mating, while the virgin females of the same age were used as controls." The age of the insects, experimental conditions, and specific response variables need clarification.

7) Zhang et al. claim that MAG extracts demonstrate the role of seminal fluid proteins in the PMR. However, MAG extracts contain more than just seminal fluid proteins. This conclusion should be softened to acknowledge this limitation.

8) The legends for some plots incorrectly state that "the small circles... denote total number of animals." Each circle likely represents an observation (e.g., the number of eggs laid by an individual female in Figure 1B). This needs correction.

9) Some legends are complete sentences (e.g., "The reproductive system of the male brown planthopper includes..."), while others are descriptive phrases (e.g., "Numbers of eggs laid per female"). The legends should consistently use concise, descriptive phrases.

10) Several methodological details currently appear in the Results section. For example: "Proteomic analysis was performed using Label-free... Production data was searched against brown planthopper MAG transcriptome databases using the Proteome Discoverer 2.2 (PD2.2, Thermo)." These details belong in the Methods section.

11) The Results section mentions the use of Bowtie for read alignment, but the Methods section specifies HISAT2. Additionally, the Results refer to a de novo transcriptome assembly, which is not described in the Methods section. These inconsistencies need resolution.

12) The subsection "Transcriptome and proteome analysis of male accessory glands" is confusing and incomplete. Key data points, such as the number of transcripts/genes transcribed in the MAGs and the proportion translated into proteins, are missing. After revision, this content may fit better within the "Identification of seminal fluid proteins" subsection.

13) The criteria for predicting seminal fluid proteins from the data are missing. Additionally, there is no comparison between the proteins identified in this study and those reported by Yu et al. (2016). This comparison would strengthen the manuscript.

14) The authors conclude that "neither MIP nor MIPR is required for increased post-mating oviposition." While this is likely, the lack of gene knockout experiments weakens this conclusion. It should be softened accordingly.

Discussion

15) Apart from SP and DUP99, other influential seminal fluid proteins of D. melanogaster (e.g., ovulin) should be included in the discussion to provide broader context.

M&M

16) The Methods section is incomplete, making it impossible to replicate the experiments or analyses. Essential details must be included.

17) The Methods subsections should be reordered to match the sequence in which the results are presented. For instance, placing the "Behavior Assays" subsection between "Mass Spectrometry" and "Proteome Analysis" is confusing and should be changed.

**Have all data underlying the figures and results presented in the manuscript been provided?**

Reviewer #1: **No: ** there are several places where data are not completely shown as addressed in the comments to authors.

Reviewer #2: Yes

Reviewer #3: **No: ** It may not be necessary, but data underlying bar plots were not provided as supporting information tables.

PLOS authors have the option to publish the peer review history of their article (what does this mean? ). If published, this will include your full peer review and any attached files.

**Do you want your identity to be public for this peer review?** For information about this choice, including consent withdrawal, please see our Privacy Policy .

Reviewer #1: No

Reviewer #2: No

Reviewer #3: **Yes: ** Juan Hurtado

**Figure resubmission:**
---

## [Decision Letter · Decision Letter 1]

2 Apr 2025

PGENETICS-D-24-01238R1

A novel male accessory gland peptide reduces female post-mating receptivity in the brown planthopper

PLOS Genetics

Dear Dr. Wu,

Thank you for submitting your manuscript to PLOS Genetics. After careful consideration, we feel that it has merit but does not fully meet PLOS Genetics's publication criteria as it currently stands. Therefore, we invite you to submit a revised version of the manuscript that addresses the points raised during the review process.

Please submit your revised manuscript within 30 days May 02 2025 11:59PM. If you will need more time than this to complete your revisions, please reply to this message or contact the journal office at plosgenetics@plos.org. Please include the following items when submitting your revised manuscript:

We look forward to receiving your revised manuscript.

Kind regards,

Takaaki Daimon

Academic Editor

PLOS Genetics

Pablo Wappner

Section Editor

PLOS Genetics

Aimée Dudley

Editor-in-Chief

PLOS Genetics

Anne Goriely

Editor-in-Chief

PLOS Genetics

**Additional Editor Comments:**

This revised manuscript has been evaluated by original reviewers.

In light of their comments, I would like to request a final round of revision.

As pointed out by Reviewer #3, please reduce the use of abbreviations and prioritize those that appear frequently to improve the readability of the manuscript.

Additionally, numerical data that underlies graphs or summary statistics should be provided in spreadsheet form as supporting information. Please submit a new supplementary table that includes numerical data for Figs. 1, 3, 4, 5, S5, S8, S10, and S13.

**Journal Requirements:**

Please ensure that the funders and grant numbers match between the Financial Disclosure field and the Funding Information tab in your submission form. Note that the funders must be provided in the same order in both places as well.

**Reviewers' comments:**

Reviewer's Responses to Questions

**Comments to the Authors:**

Reviewer #2: Dear authors and editor,

I have just finished reading the other reviewers' corrections, almost all of which have been adopted by the authors, including my own. Approximately 25 articles that should be cited have been added, and the materials and methods have been considerably expanded to make the methodologies reproducible.

The authors have done extensive work improving the manuscript, taking into account the considerations of the three reviewers. Therefore, I consider it worthy of publication.

Reviewer #3: I thoroughly enjoyed reading this interesting manuscript. The authors have done an excellent job addressing the reviewers' comments from the initial review. I now have no major concerns about the quality of this manuscript and am happy to recommend it for publication. However, I do have some minor comments that should be straightforward to address:

1. Abbreviations: The manuscript currently contains a large number of abbreviations, which can make it difficult for readers to follow. I suggest either reducing the number of abbreviations or including a table of abbreviations for reference. Below is a list of the abbreviations used throughout the manuscript:

PMR: post-mating response

SP: sex peptide

MAG: male accessory glands

macc: maccessin

SPR: sex peptide receptor

MIP: myoinhibitory peptide

MIPR: myoinhibitory peptide receptor

HP-1: head peptide

SPF-L: Selenoprotein F-Like

ITPL-1: ion transport peptide-like

SFP: seminal fluid protein

ITPa: recombinant amidated ITP

BPH: brown planthopper

sNPF: short neuropeptide F

NPYLR1: short neuropeptide F receptor

2. Line 188: Please add "the" before "Mexican fruit fly" for grammatical correctness.

3. Line 189: Replace "we" with "our findings" or "our results".

4. Figures 1A, 4A, 5A, and S5A: The vertical axis title "Proportion of females courted" could be misleading, as it may imply that female attractiveness was assessed rather than receptivity. Consider revising it to "Proportion of females accepting courting males" or something similar to clarify the intended meaning.

5. Line 226: Add "brown planthopper" before "(BPH)" because the abbreviation was first introduced here.

6. Lines 233–244: To improve flow and clarity, consider explicitly stating whether a de novo or genome-guided assembly was performed before describing the sequence similarity searches of the assembled transcripts. This would help connect the pre- and post-assembly parts of this paragraph.

7. Lines 329–330: The numbers 373 and 209 do not match the values shown in Figure S3B (394 and 205, respectively). Please revise these numbers for accuracy.

8. Lines 331–332: The SignalP column is missing from Table S2. Please ensure this column is included to provide complete information.

9. Lines 340–342: In addition to macc, Table S2 includes nearly a dozen other unknown SFP genes with higher expression levels in the MAGs and detected peptide fragments. It would be helpful to provide a more detailed explanation of why macc was chosen for further exploration over these other genes. For example, was it selected because it was the only one with a predicted signal peptide, or due to its annotation as an ion transport peptide precursor? Also, please clarify whether macc was included in the list of putative seminal fluid proteins or seminal fluid protein candidates.

10. Line 756: Please clarify the type of assembly performed using Trinity software (e.g., de novo or genome-guided) and include relevant run parameters to provide a complete description of the methods.

**Have all data underlying the figures and results presented in the manuscript been provided?**

Reviewer #2: Yes

Reviewer #3: **No: ** It may not be necessary, but some data e.g., underlying bar plots were not provided as supporting information tables.

PLOS authors have the option to publish the peer review history of their article (what does this mean? ). If published, this will include your full peer review and any attached files.

**Do you want your identity to be public for this peer review?** For information about this choice, including consent withdrawal, please see our Privacy Policy .

Reviewer #2: No

Reviewer #3: **Yes: ** Juan Hurtado

**Figure resubmission:**
---

## [Editor Report · Decision Letter 2]

23 Apr 2025

Dear Dr Wu,

We are pleased to inform you that your manuscript entitled "A novel male accessory gland peptide reduces female post-mating receptivity in the brown planthopper" has been editorially accepted for publication in PLOS Genetics. Congratulations!

Yours sincerely,

Takaaki Daimon

Academic Editor

PLOS Genetics

Pablo Wappner

Section Editor

PLOS Genetics

Aimée Dudley

Editor-in-Chief

PLOS Genetics

Anne Goriely

Editor-in-Chief

PLOS Genetics

Comments from the reviewers (if applicable):

The authors have done a nice job revising the manuscript.

I think it is now ready for publication.

**Data Deposition**

http://datadryad.org/submit?journalID=pgenetics&manu=PGENETICS-D-24-01238R2

**Press Queries**

---

## [Editor Report · Acceptance letter]

PGENETICS-D-24-01238R2

A novel male accessory gland peptide reduces female post-mating receptivity in the brown planthopper

Dear Dr Wu,

We are pleased to inform you that your manuscript entitled "A novel male accessory gland peptide reduces female post-mating receptivity in the brown planthopper" has been formally accepted for publication in PLOS Genetics! Your manuscript is now with our production department and you will be notified of the publication date in due course.

With kind regards,

Zsofia Freund

PLOS Genetics

On behalf of:
